# Functional insights from a surface antigen mRNA-bound proteome

**Larissa Melo do Nascimento[1], Franziska Egler[1], Katharina Arnold[1], Nina Papavasiliou[2], Christine Clayton[1], Esteban Erben[1,2]†***

[1]Centre for Molecular Biology of Heidelberg University (ZMBH), Heidelberg, Germany; [2]Division of Immune Diversity, Deutsche Krebsforschungszentrum (DKFZ), Heidelberg, Germany

**Abstract** *Trypanosoma brucei* is the causative agent of human sleeping sickness. The parasites' variant surface glycoprotein (VSG) enables them to evade adaptive immunity via antigenic variation. VSG comprises 10% of total cell protein and the high stability of VSG mRNA is essential for trypanosome survival. To determine how VSG mRNA stability is maintained, we used mRNA affinity purification to identify all its associated proteins. CFB2 (cyclin F-box protein 2), an unconventional RNA-binding protein with an F-box domain, was specifically enriched with VSG mRNA. We demonstrate that CFB2 is essential for VSG mRNA stability, describe cis acting elements within the VSG 3'-untranslated region that regulate the interaction, identify trans-acting factors that are present in the VSG messenger ribonucleoprotein particle, and mechanistically explain how CFB2 stabilizes the mRNA of this key pathogenicity factor. Beyond *T. brucei*, the mRNP purification approach has the potential to supply detailed biological insight into metabolism of relatively abundant mRNAs in any eukaryote.

***For correspondence:**
eerben@iib.unsam.edu.ar

**Present address:** †Instituto de Investigaciones Biotecnológicas, Universidad Nacional de San Martín, Provincia de Buenos Aires, Argentina

## Introduction

In the cytosol, eukaryotic mRNAs interact with numerous proteins that influence mRNA folding, localization, translation efficiency, and longevity (*Khong and Parker, 2020*). The resulting messenger ribonucleoprotein (mRNP) contains, on average, about 15 proteins (and sometimes also regulatory RNAs) which may have competing activities and change during the lifetime of the mRNA (*Khong and Parker, 2020*). The behaviour of individual transcripts in vivo is therefore the result of dynamic interactions between the various mRNP components and can only be understood if the identities of those components are known. However, although there are now numerous papers that take a protein-centric approach – cataloguing the mRNAs that are bound to specific proteins – cataloguing the components of individual mRNPs has proven to be extremely challenging, mainly because of the very low molar ratio of mRNAs to proteins in cells. Consequently, so far there has been only one report of successful purification of native eukaryotic mRNPs, which characterized the proteins bound to two abundant GFP reporter mRNAs in the nematode *Caenorhabditis elegans* (*Theil et al., 2019*).

*Trypanosoma brucei* (*T. brucei*) is a unicellular eukaryotic parasite that causes lethal diseases, including human sleeping sickness, in sub-Saharan Africa. *T. brucei* and related 'salivarian' trypanosomes are transmitted between mammals by tsetse flies, proliferating in the midgut as the 'procyclic' form. Within mammals, 'bloodstream-form' *T. brucei* multiply in the blood and tissue fluids. If untreated, human sleeping sickness is almost always lethal, while infections of domestic animals with salivarian trypanosomes cause immense economic damage. The infections can last months or years, because *T. brucei* escapes the host immune system by periodically changing its highly antigenic surface coat.

The genome organization of trypanosomes and related organisms is unusual, since protein-coding genes lack individual promoters. Instead, they are arranged in long polycistronic transcription units that are constitutively transcribed by RNA polymerase II. Individual mRNAs are co-transcriptionally excised: a capped 5'-end is created by *trans* splicing of a 39 nt capped 'spliced leader' RNA, and the 3'-end is generated by cleavage and polyadenylation (*Clayton, 2019*). As a consequence, trypanosomes rely heavily on post-transcriptional mechanisms for control of gene expression (*Clayton, 2019*). Abundant mRNAs are usually very stable and are often transcribed from multiple gene copies. Differential regulation is effected by numerous sequence-specific RNA-binding proteins that influence mRNA processing, translation, and decay (*Clayton, 2019*; *Lueong et al., 2016*). The parasites also have six different isoforms of the cap-binding translation initiation factor eIF4E (EIF4E1-6), five different versions of its partner eIF4G (EIF4G1-5), and two versions of poly(A)-binding protein (PABP1 and PABP2) (*Zoltner et al., 2018*). In various combinations, these are implicated in either enhancing or suppressing translation (*Freire et al., 2017*; *Terrao et al., 2018*).

This surface of salivarian trypanosomes is composed of ~11 million variant surface glycoprotein (VSG) molecules that are mono-allelically expressed from specialized telomeric expression sites. In *T. brucei*, each expression site consists of an RNA polymerase I promoter (*Günzl et al., 2003*; *Kooter and Borst, 1984*) followed by several different 'expression site-associated genes' (*ESAG*s), some repetitive sequences, the *VSG* gene, and finally, telomeric repeats (*Hutchinson et al., 2016*; *Hertz-Fowler et al., 2008*; *Müller et al., 2018*). Although there are at least 10 alternative expression sites, all but one are suppressed by epigenetic mechanisms. Antigenic variation is effected through a combination of expression site transcription switching, and genetic rearrangements that replace the currently expressed *VSG* gene with a different one from a repertoire of alternative *VSG* genes located at telomeres or in sub-telomeric arrays (*Hall et al., 2013*; *Jackson et al., 2012*; *Faria et al., 2019*).

Production of functional mRNAs by RNA polymerase I is possible in trypanosomes because the cap, which is necessary for translation, is added by *trans* splicing. For *VSG*, polymerase I transcription is also necessary in order to produce large amounts of mRNA from a single gene: *VSG* mRNA comprises at least 5% of total mRNA, and RNA polymerase I transcription from *VSG* promoters is at least 10 times more efficient than transcription of chromosome-internal regions RNA polymerase II (*Clayton, 2019*). Moreover, association of the active *VSG* expression site with the site of spliced leader RNA production results in extremely efficient processing of the *VSG* mRNA (*Faria et al., 2020*). However, these measures alone do not suffice: the *VSG* mRNA is also extremely stable, having a half-life of 1–2 hr compared with less than 20 min for most other mRNAs, including *ESAG*s (*Ehlers et al., 1987*; *Vanhamme et al., 1999*; *Ridewood et al., 2017*; *Fadda et al., 2014*). This stability has been shown to require a 16-mer sequence within the VSG 3'-untranslated region (3'-UTR), which is common to these otherwise highly variant genes (*Ridewood et al., 2017*; *Berberof et al., 1995*). After uptake into the tsetse fly vector, as the trypanosomes convert to the procyclic form, *VSG* transcription is shut off, the mRNA becomes unstable (*Ehlers et al., 1987*; *Berberof et al., 1995*), and the VSG coat is lost.

In this work, we aimed to determine the mechanism by which *VSG* mRNA stability is maintained, by identifying proteins that are specifically associated with the *VSG* mRNP. As a control, we studied the mRNA encoding alpha tubulin (*TUB*), which is transcribed by RNA polymerase II from at least 19 tandemly repeated genes (*Ersfeld et al., 1998*), is also relatively stable, and comprises about 1.5% of the RNA (*Fadda et al., 2014*). Through sequential purification of the two mRNPs, we were able to identify not only proteins that were in both mRNPs, and some that were specific to either *VSG* or *TUB*. This enabled us to identify CFB2 (cyclin F-box protein 2), a protein that specifically stabilizes *VSG* mRNA through recruitment of a stabilizing, translation-promoting complex.

## Results

### The *VSG* mRNP proteome

To investigate the mechanism by which *VSG* mRNA is stabilized in bloodstream forms, we purified the *VSG* mRNP from cells expressing one particular VSG: VSG2. We, like *Theil et al., 2019*, adapted methods previously applied to capture proteins associated with the *Xist* noncoding RNA (*McHugh et al., 2015*; *Chu et al., 2015*; *Minajigi et al., 2015*). After UV cross-linking of proteins to

RNA, we used biotinylated 90-mers complementary to the entire *VSG* mRNA (*Supplementary file 1*) to purify the *VSG* and *TUB* mRNPs (*Figure 1A*). The two mRNAs were purified sequentially, in alternating order, from the same cell extracts (*Figure 1A*) to make the background for each as similar as possible; in principle it should be possible to add a third or even a fourth purification to the cascade. At least 1000-fold purification relative to rRNA was obtained (*Figure 1B*), suggesting final mRNA: rRNA ratios of ~1:1 (*Haanstra et al., 2008*). Quantitative mass spectrometry detected 664 different proteins with at least two peptides (*Supplementary file 2*, sheet 5). 97 proteins were found in all three *VSG* replicates, and 18 in all three *TUB* replicates; this discrepancy probably reflects the relative abundances of the mRNAs in the original lysates. RNA-binding-domain proteins associated with

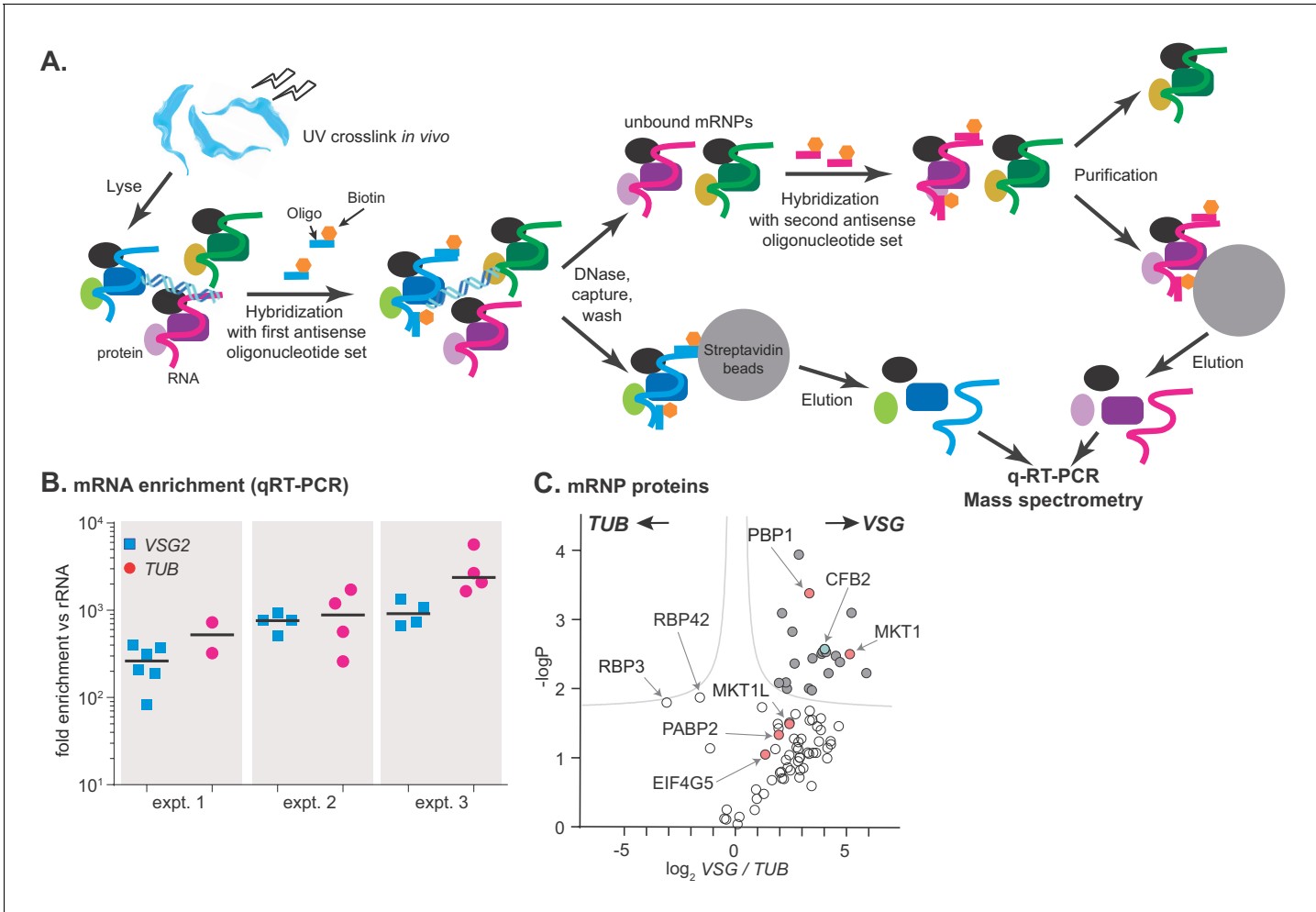

**Figure 1.** RNA antisense purification identifies proteins that interact directly with variant surface glycoprotein (VSG) mRNA. (**A**) A schematic overview of the method. Bloodstream-form *Trypanosoma brucei* were subjected to UV irradiation. After cell lysis, the lysate was incubated with streptavidin-coated magnetic beads. The unbound portion was then incubated with biotinylated 90-nt-long DNA probes, to hybridize to either alpha-tubulin (TUB) or VSG2 mRNA, then probe-target complexes were captured by streptavidin-coated magnetic beads. To decrease background, samples were treated with DNase I. The supernatant was collected, and the second set of ribonucleoprotein complexes (VSG2 or TUB) was captured in a similar way. Protein and RNA were eluted from the beads, and subjected to mass spectrometry (LC-MS/MS) for protein identification and real-time quantitative PCR (RT-qPCR) for relative RNA quantification. (**B**) Enrichment of VSG2 and TUB transcripts after RNA antisense purification. The individual data points show RNA levels relative to rRNA for each independent pull-down, measured by RT-qPCR, and bars represent means. Experiment 1 includes some preparations that were purified using only VSG. Before mass spectrometry of VSG or TUB preparations from each experiment, the material for all pull-downs shown was pooled. (**C**) The triplicate experiments identified proteins that reproducibly enriched with VSG2 relative to TUB and vice versa. Proteins significantly enriched (FDR 1%; s0 = 0.1) are filled dark grey. Proteins associated with the MKT1 complex are in pink and CFB2 (cyclin F-box protein 2) is cyan. The data for this graph are in *Supplementary file 2*, sheet 2.

The online version of this article includes the following figure supplement(s) for figure 1:

**Figure supplement 1.** CFB2 (cyclin F-box protein 2) domain conservation and evolution.

both mRNAs included DRBD18 (*Lott et al., 2015*), RBP3 (*Wurst et al., 2009*), UBP2 (*Hartmann et al., 2007*), DRBD3/PTB1 (*Estévez, 2008*; *Stern et al., 2009*), and RBP42 (*Das et al., 2012*). Interestingly, RBP3 was significantly more enriched with *TUB* than with *VSG* (*Supplementary file 2*, sheets 3 and 5). RBP3 is essential for normal bloodstream-form trypanosome growth but a previous microarray analysis did not detect the RBP3-*TUB* mRNA association (*Wurst et al., 2009*). RBP42, which is a polysome-associated protein that preferentially binds to coding regions (*Das et al., 2012*), also preferred *TUB*. Notably, only one of the two poly(A)-binding protein orthologues, PABP2, was reproducibly associated with either *VSG* or *TUB* mRNAs, supporting previously suggested (*Zoltner et al., 2018*) specialized roles for PABP2 and its orthologue PABP1.

Up to 43 proteins were significantly enriched in the *VSG2* mRNP, depending on the criteria applied (*Figure 1C*, *Supplementary file 2*, sheet 1). Of these, all but three were detected in the total poly(A)+ mRNP proteome (*Lueong et al., 2016*), supporting direct binding to mRNA in vivo. The proteins with general mRNA-related functions – PABP2, helicases, translation factors, and ribosomal proteins – were probably enriched because *VSG* mRNA is more abundant than *TUB*. *VSG* mRNA-associated proteins with known RNA-binding domains were ALBA3, ZC3H28, ZC3H32, ZC3H41. ALBA3 has previously been implicated in translation enhancement and developmental regulation in procyclic forms of the parasite (*Mani et al., 2011*; *Subota et al., 2011*), and ZC3H41 is associated with SL RNA, the precursor for mRNA *trans* splicing (*Eliaz et al., 2017*). Although ZC3H32 is bloodstream-form-specific and essential, a tagged version showed no evidence of specificity in mRNA binding (*Klein et al., 2017*). The roles of ZC3H28 and ZC3H41 are not known.

To find candidates for stabilization of *VSG* mRNA, we focused on proteins that were highly enriched in the *VSG* mRNP, are known to bind well to mRNA (*Lueong et al., 2016*), and are expressed only in the bloodstream form (*Fadda et al., 2014*; *Siegel et al., 2010*; *Jensen et al., 2014*; *Dejung et al., 2016*). We also chose proteins that are capable of increasing mRNA stability or translation when artificially 'tethered' to an mRNA (*Lueong et al., 2016*; *Erben et al., 2014*). In this assay, we express the protein of interest as a fusion with the lambdaN peptide, using cells that also express a reporter mRNA that contains five copies of the boxB sequence, which binds the lambdaN peptide with very high affinity. One possible candidate was ERBP1, but this shows only moderate developmental regulation; its co-purification with the *VSG* mRNP may be linked to its association with the endoplasmic reticulum (*Bajak et al., 2020*). The remaining candidate was CFB2 (Tb927.1.4650).

## CFB2 is associated with *VSG* mRNA

The *CFB2* gene is downstream of several genes encoding CFB1, a related protein (*Benz and Clayton, 2007*). Both *CFB1* and *CFB2* mRNAs are much more abundant in bloodstream forms than in procyclic forms, but mass spectrometry (*Dejung et al., 2016*) and ribosome profiling results (*Jensen et al., 2014*; *Antwi et al., 2016*) suggest that CFB2 predominates in bloodstream forms. *CFB2* mRNA persists in stumpy-form trypanosomes (*Silvester et al., 2018*; *Naguleswaran et al., 2018*), which are growth-arrested VSG-expressing bloodstream forms that are poised for differentiation to the procyclic form. Within the tsetse fly, *CFB2* mRNA is present only in forms that express VSG (*Telleria et al., 2014*; *Savage et al., 2016*; *Vigneron et al., 2020*), whereas *CFB1* mRNA is also present in the epimastigote form, which lacks VSG (*Vigneron et al., 2020*). *CFB* genes are trypanosome-specific; at least one *CFB* gene is present in all Trypanosoma, but they are absent in Leishmania. Alignments of the different regions of the protein suggest that different copies, where present, arose by duplication after species divergence (*Figure 1—figure supplement 1*). It is notable that whereas all Salivaria have several *CFB* genes, the intracellular Stercoraria have only one. Moreover, it was already known that depletion of CFB2 from bloodstream forms caused rapid G2 arrest (*Benz and Clayton, 2007*) – a phenotype that was also seen after RNAi targeting *VSG* mRNA (*Sheader et al., 2005*).

To confirm association of CFB2 with *VSG* mRNA, we integrated boxB loops into the actively expressed *VSG* gene immediately after the *VSG2* stop codon, preserving the endogenous *VSG* 3'-UTR and polyadenylation site (*Figure 2A*). We co-expressed a chimeric GFP protein bearing a streptavidin-binding peptide at the C-terminus and, at the N-terminus, the lambdaN peptide (N-GFP-SBP) (*Figure 2A,B*); expression was tetracycline-inducible. In addition, 6x-myc-CFB2 was constitutively expressed in the same cells. Affinity purification on a streptavidin matrix allowed us to pull down the *VSG-boxB* mRNA using N-GFP-SBP. Detection of Myc-CFB2 in the pull-down depended

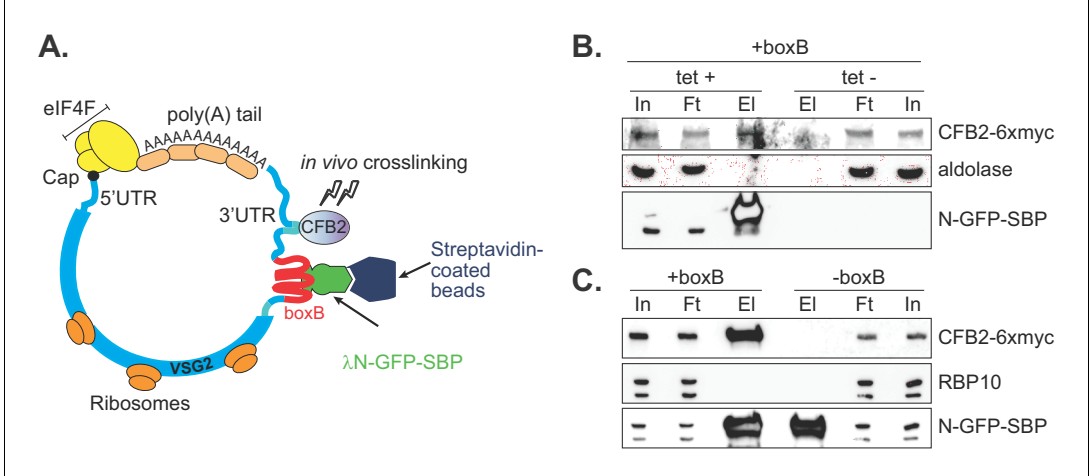

**Figure 2.** CFB2 (cyclin F-box protein 2) associates with VSG2 mRNA. (A) Method used. The lambda-GFP-SBP binds to boxB aptamer-tagged *VSG2* mRNAs via the lambda moiety located at the N-terminal and interacts with immobilized streptavidin via the SBP moiety at the C-terminus. (B) CFB2-myc is pulled down with boxB-containing *VSG2* mRNA. Bloodstream cells expressing boxB-tagged *VSG2* mRNA from the active expression site, CFB2-6xmyc from the *RRNA* locus, and tetracycline-inducible lambdaN-GFP-SBP were grown; one culture was treated with tetracycline for 6 hr (+tet) and the other was not (-tet). The lambdaN-GFP-SBP, with associated mRNA and protein, was purified from cell lysates (25 mg of the total protein). For Western analysis, 85% of the eluates (El; corresponding to ~21 mg of input protein) and 40 μg total input (In) or flowthrough (Ft) proteins were resolved by SDS-PAGE and analysed by Western blotting using specific antibodies (eluate: flowthrough loading of 500:1). Panel B shows that pull-down of CFB2-6xmyc was dependent on the presence of lambdaN-GFP-SBP. Raw data for this figure are in *Figure 2—figure supplement 1*. (C) This is similar to B, except that all cells expressed lambdaN-GFP-SBP, but one line did not have boxB sequences in the *VSG* mRNA. This shows that pull-down of CFB2-6xmyc by lambdaN-GFP-SBP depended on the boxB sequences.

The online version of this article includes the following figure supplement(s) for figure 2:

**Figure supplement 1.** Effects of CFB2 (cyclin F-box protein 2) RNAi and uncropped images for *Figure 2* and *Figure 4*.

on the presence of both lambda-GFP-SBP (*Figure 2B*) and the *VSG2*-associated *boxBs* (*Figure 2C*, *Figure 2—figure supplement 1*). Other abundant proteins such as RBP10 (*Wurst et al., 2012*; *Mugo and Clayton, 2017*) and aldolase (*Clayton, 1987*) were not enriched (*Figure 2B,C*). In a converse experiment, we used cells expressing CFB2-6xmyc but no boxB reporter. After immunoprecipitation, *VSG2* mRNA was enriched 1.8- to 3.8-fold relative to rRNA in 4/5 experiments (*Figure 2—figure supplement 1*).

## The MKT1-PBP1 complex is associated with *VSG* mRNA

CFB1 and CFB2 have three possible functional domains: the cyclin F-box, a conserved central region, and the MKT1 interaction motif (*Figure 3A*). Cyclin F-box proteins interact with SKP1 and form an E3 ubiquitin ligase. The residues required for the interaction in Opisthokonts (*Bai et al., 1996*; *Schulman et al., 2000*; *Chan et al., 2013*) are conserved in CFB2 (*Figure 3B*). The interaction of CFB2 with *T. brucei* SKP1 was confirmed in a yeast two-hybrid assay (*Figure 3C*) and was prevented by mutation of critical F-box residues (*Figure 3—figure supplement 1*).

The C-terminus of CFB2 includes a motif, RYRHDPY, which is required for the interaction of CFB2 with MKT1 (*Singh et al., 2014*; *Melo do Nascimento et al., 2020*; *Figure 3—figure supplement 1*). As noted in the Introduction, *T. brucei* MKT1 forms a complex with PBP1, LSM12, XAC1, and PABP2 (*Singh et al., 2014*; *Melo do Nascimento et al., 2020*), and also preferentially recruits one of the six alternative cap-binding translation initiator factor complexes, EIF4E6-EIF4G5 (*Singh et al., 2014*; *Freire et al., 2017*; *Figure 3D*). Recruitment of the MKT complex stabilizes bound mRNAs and promotes translation (*Singh et al., 2014*). Evidence so far suggests that although both MKT1 and PBP1 have some intrinsic RNA-binding activity, they are recruited to specific mRNAs by various different RNA-binding proteins, resulting in enhanced mRNA abundance and translation (*Lueong et al., 2016*; *Singh et al., 2014*; *Liu et al., 2020*). CFB2 showed clear co-purification with MKT1 and XAC1 (*Singh et al., 2014*; *Melo do Nascimento et al., 2020*). Correspondingly, both MKT1 and PBP1 were highly enriched in the *VSG* mRNP (*Figure 1C*, *Supplementary file 2*, sheet 1). EIF4G5, EIF4E6,

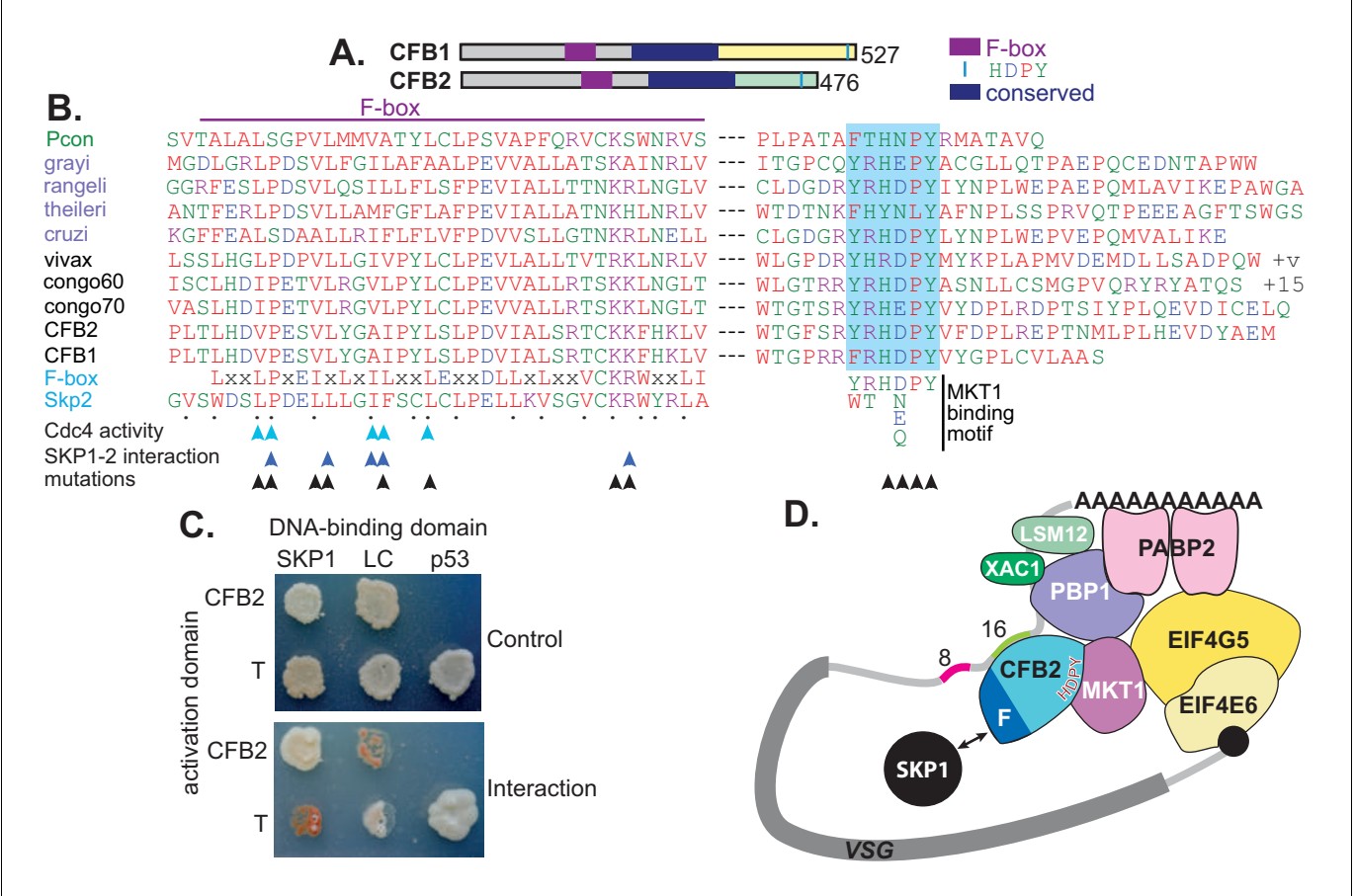

**Figure 3.** CFB2 (cyclin F-box protein 2) has three conserved domains. (A) Schematic depiction of *Trypanosoma brucei* CFB1 and CFB2. Similar sequences outside the F-box and HDPY region are in grey and regions conserved in *Trypanosoma* and Paratrypanosoma are in other colours as indicated. (B) Sequence alignments created in ClustalOmega. The F-box is on the left and the region with C-terminal HNPY (highlighted in blue) on the right. The colour code for residues is red: non-polar; green: polar; blue: acidic; purple: basic. The Paratrypanosoma sequence (green label) is at the top. Stercoraria (*Trypanosoma* grayi, Trypanosoma rangeli, *Trypanosoma* theileri, and *Trypanosoma cruzi*) are labelled in mauve and Salivaria are in black (*Trypanosoma vivax*, *Trypanosoma* congolense, and *T. brucei* CFB1 and CFB2). Most organisms shown have several very similar CFB gene copies; only one is shown unless sequences are different. '+v' indicates variable C-termini of different *T. vivax* orthologues; '+15' indicates 15 more amino acid residues. At the bottom are a mammalian F-box consensus ('F-box') and human SKP2. The points below SKP2 indicate residues implicated in its interaction with SKP1 (*Schulman et al., 2000*). Cyan arrows indicate residues required for yeast Cdc4 activity (*Bai et al., 1996*) and dark blue arrows the residues required for the human SKP1-SKP2 interaction (*Chan et al., 2013*). Black arrows indicate residues mutated to alanine in our studies. Phosphorylation of CFB2 was detected at tyrosine 431, 16 residues N-terminal to the HDPY signal (*Urbaniak et al., 2013*). (C) The F-box of CFB2 interacts with SKP1: yeast two-hybrid assay. Fusions of SKP1, LaminC (negative control), and p53 with the DNA-binding domain were co-expressed with CFB2 or SV40 T-antigen fused with the transcription activation domain. The interaction between SV40 T-antigen and p53 is the positive control. The upper panel shows selection of transformants on 'double drop-out' medium and the lower panel shows selection for the interaction on 'quadruple drop-out' medium. (D) A model for CFB2 function. The HDPY motif of CFB2 interacts with MKT1, which recruits PBP1 and EIF4G5. PBP1 recruits PABP2, XAC1, and LSM12, and EIF4G5 is complexed with EIF4E6. Meanwhile, the F-box of CFB2 interacts with SKP1; this might, or might not, be compatible with MKT1 and mRNA binding. CFB2 is shown bound to the 16-mer; some evidence to support this hypothesis is presented later.

The online version of this article includes the following figure supplement(s) for figure 3:

**Figure supplement 1.** Roles of the F-box and HDPY sequence in protein-protein interactions.

and EIF4G4 were also detected exclusively with VSG, but the cap-binding partner of EIF4G4, EIF4E3, was not detected. We therefore hypothesized that recruitment of CFB2 to the *VSG* mRNA results in cooperative assembly of an mRNP that includes the MKT complex, PABP2 and EIF4E6-EIF4G5 (*Figure 3D*).

## Depletion of CFB2 causes selective loss of *VSG* mRNA

To test our hypothesis, we first examined the effects of CFB2 depletion. We had previously shown that RNAi-mediated depletion of CFB2 resulted in almost immediate G2 arrest, with an accumulation of flagella and basal bodies (*Benz and Clayton, 2007*). To find out whether this was accompanied by a specific reduction in *VSG* mRNA, we induced *CFB2* RNAi and measured mRNAs by RT-PCR 3 and 6 hr later. TUB mRNA was unaffected but *VSG* mRNA was already reduced within 3 hr of *CFB2* RNAi induction (*Figure 4A,B*). Importantly, preliminary data showed that this was also true for a cell line expressing *VSG3* followed by the 3'-UTR of *VSG2* (*Figure 2—figure supplement 1*), and transient transfection of *CFB2* double-stranded RNA into cells expressing VSG222 also caused *VSG222* mRNA reduction (*Figure 2—figure supplement 1*), suggesting that the *CFB2* RNAi effect does not

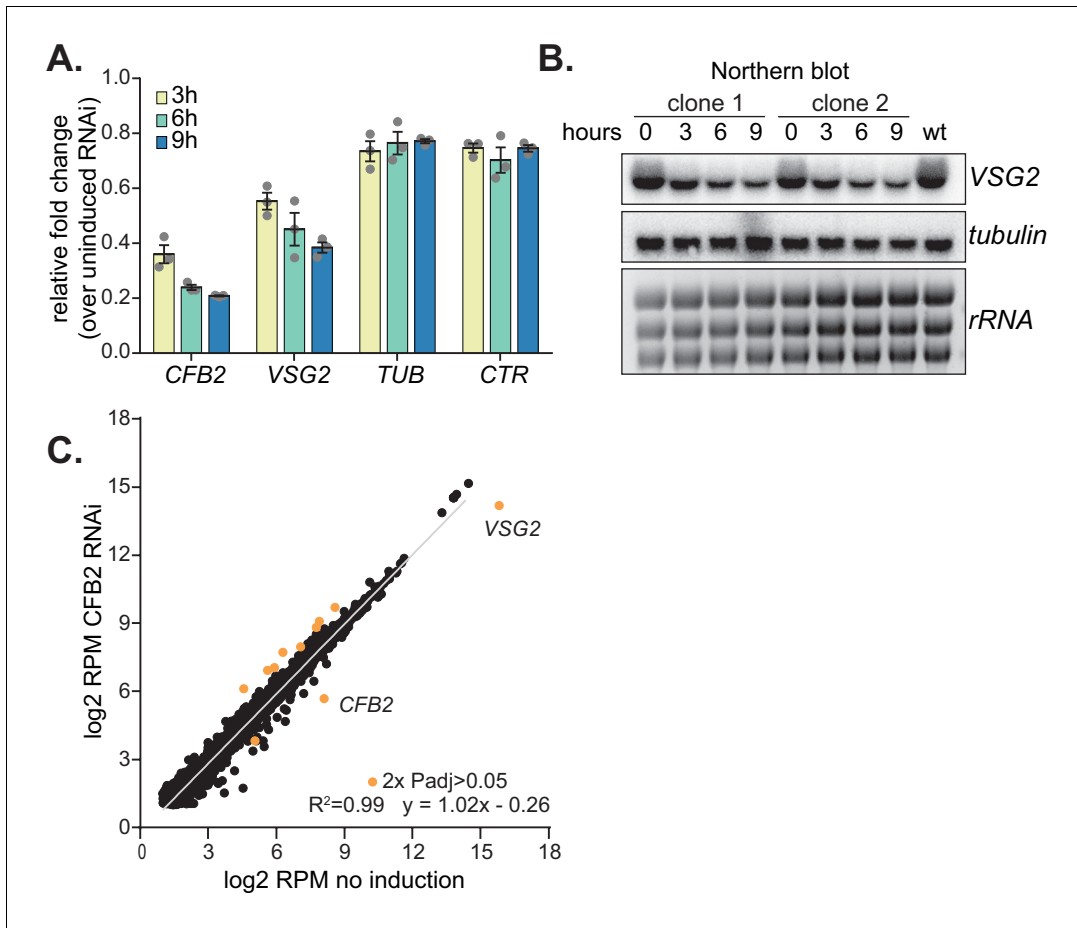

**Figure 4.** Depletion of CFB2 (cyclin F-box protein 2) results in selective loss of variant surface glycoprotein (VSG) mRNA. (**A**) Relative quantification of RNA transcript levels corresponding to the active *VSG2* and alpha-tubulin (*TUB*) mRNAs detected at different sampling time-points (3, 6, 9 hr) after induction of CFB2 RNAi. CTR (co-transposed region) is a sequence located upstream of the VSG gene, which is transcribed from the same promoter but present only in the mRNA precursor. Results were derived from three biological replicate experiments with standard deviation indicated with error bars, normalized against 18S. (**B**) Representative Northern blot analysis of transcript levels after depletion of CFB2 in cells expressing VSG2; details as in (**A**). Raw data for this figure are in *Figure 2—figure supplement 1*. (**C**) Effect of *CFB2* RNAi on the transcriptome. Cells with tetracycline-inducible *CFB2* RNAi were used. The average reads per million (RPM) from two replicates were plotted for each open reading frame. Values below two were excluded. Results for Lister427 coding regions are shown (*Supplementary file 3*). Data for genes which gave an RPM ratio of more than two-fold increase or decrease, and a DeSeq2 adjusted p-value of less than 0.05, are highlighted in orange.

The online version of this article includes the following figure supplement(s) for figure 4:

**Figure supplement 1.** Effects of CFB2 (cyclin F-box protein 2) RNAi on the transcriptome.

require a specific *VSG* coding region. Inhibition of VSG synthesis is known to induce translation arrest within 24 hr (*Smith et al., 2009*), but no general translation inhibition was observed over the first 8 hr of *CFB2* RNAi (*Figure 2—figure supplement 1*).

We next examined the effect of CFB2 reduction on the transcriptome. After 9 hr of CFB2 depletion, the only mRNAs that were significantly reduced were those encoding CFB2, VSG2, and, to a lesser extent, the mRNA from locus Tb927.8.1945, which encodes a hypothetical protein of unknown function (*Figure 4C*, *Figure 4—figure supplement 1*, *Supplementary files 3* and *4*). As previously shown (*Benz and Clayton, 2007*), *CFB2*, but not *CFB1*, transcripts decreased after RNAi induction; the minor decrease in coverage over the *CFB1* genes can be assigned to sequences that are also present in *CFB2* (*Figure 5A*). *Figure 5B* shows reads over the active VSG expression site: the active *VSG2* mRNA was decreased, while mRNAs from the co-transcribed *ESAG* genes were unaffected. This indicates that the effect on *VSG2* mRNA most likely operates at the post-transcriptional level, although we cannot rule out the possibility that some *ESAG* reads come from copies elsewhere in the genome. The level of the *VSG* pre-mRNA, as judged by the *VSG2* co-transposed region (CTR) measurement, also remained constant (*Figure 4A*).

Interestingly, 13 different mRNAs were >1.5× increased after CFB2 depletion (*Figure 4C*, *Supplementary file 4*). Their products included ESAG5-related proteins, cysteine peptidases, the surface protease MSPC (*LaCount et al., 2003*), and the invariant surface glycoprotein ISG65 (*Ziegelbauer et al., 1995*; *Chung et al., 2004*). Some other possible membrane protein mRNAs (*Shimogawa et al., 2015*) were also slightly (>1.3×), but significantly, increased. It was previously shown that when VSG synthesis is inhibited using *VSG* antisense morpholino oligonucleotides, there is some morphological distortion within the secretory pathway, but no effect on glycosyl phosphatidylinositol synthesis or vesicular transport of a lysosomal protein (*Ooi et al., 2018*). We therefore

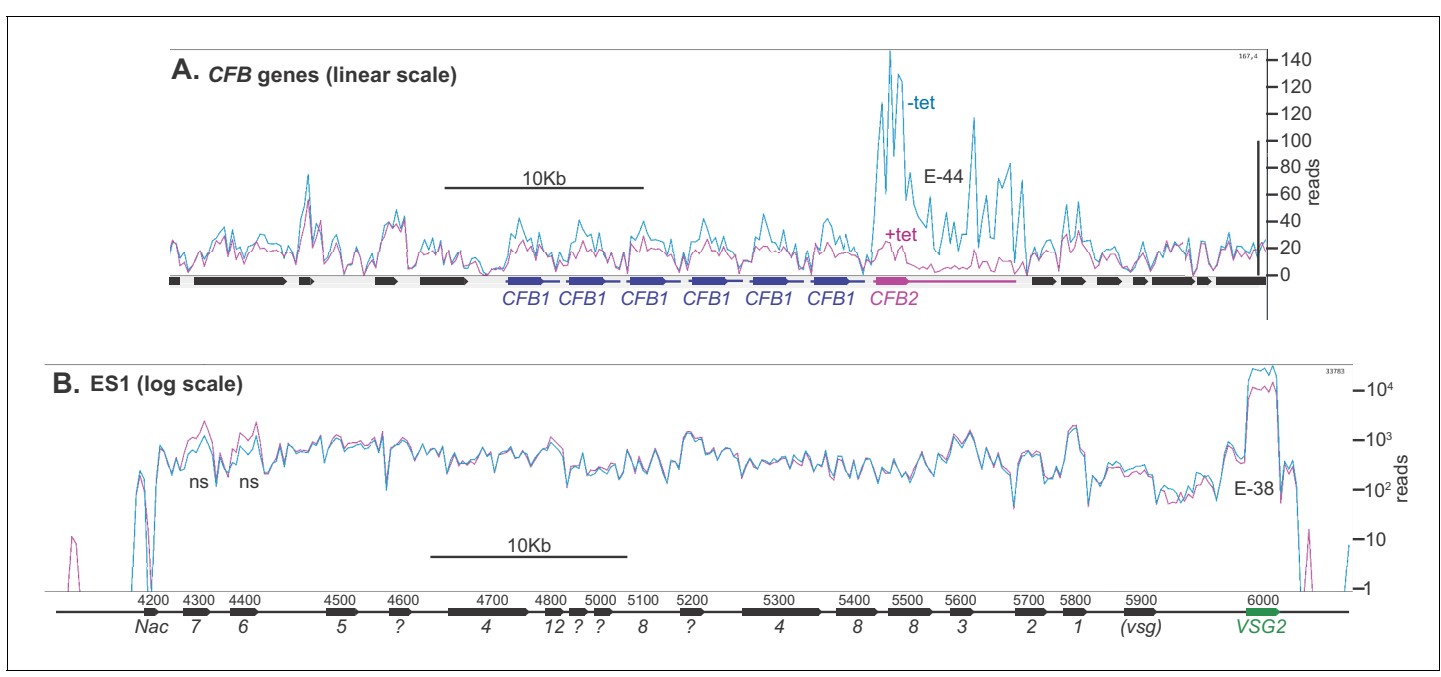

**Figure 5.** Depletion of CFB2 (cyclin F-box protein 2) does not affect expression site transcripts. Reads were aligned to the Lister427 genome, allowing one match per read, and visualized using Artemis. Reads from two samples, one with, and one without, tetracycline, were compared because they had similar overall read counts. (A) *CFB1* and *CFB2* genes, shown on a linear scale. CFB1 open reading frames are in blue and CFB2 in magenta, and the approximate extents of the untranslated regions are indicated by lines. *CFB1* and *CFB2* share some coding region sequence, but the untranslated regions are different. After tetracycline addition, some reads seen over the *CFB2* open reading frame probably actually originate from *CFB1* genes. E values are the significance of the change from DeSeq2. (B) Reads aligned over bloodstream-form expression site 1 (BES1), shown on a $\log_{10}$ scale. Most *ESAG* genes are present in multiple copies in the genome, in expression sites and elsewhere. *ESAG* mRNAs are less abundant and presumably less stable than the *VSG* mRNA. ESAGs are normally transcribed from the expression site, although some matching mRNAs may arise from copies elsewhere in the genome. The effect of *CFB2* RNAi was exclusive to VSG2. The p-values for significant differences are shown. NS, no significant differences.

speculate that loss of *VSG* mRNAs might allow mRNAs encoding other membrane proteins increased access to the secretory pathway. This might facilitate translation, which might in turn indirectly enhance mRNA stability. However, for the remaining increased mRNAs, there is no evidence for association with the secretory pathway (*Dean et al., 2017*).

## Depletion of CFB2 causes accumulation of cells containing internal flagella

Depletion of *VSG* mRNA by RNAi causes not only a G2 block but also, after 60 hr after RNAi induction, multiple internal flagella (*Sheader et al., 2005*). The effects of *CFB2* RNAi were similar, but much more rapid. Cells started to accumulate at G2/M almost immediately, with cell death commencing after about 24 hr (*Benz and Clayton, 2007*). The effects of *CFB2* RNAi are presumably fast because there are only about five *CFB2* mRNAs per cell (*Fadda et al., 2014*) and the protein is unstable (see below). 16 hr after RNAi induction, the cells had numerous flagella and basal bodies (*Benz and Clayton, 2007*). Electron microscopy revealed that at this time, nearly all cells had several external flagella, including both an axoneme and a paraflagellar rod. As seen after depletion of VSG mRNA, internal membrane and nonmembrane bound flagellum were also observed (*Figure 6*, *Figure 6—figure supplements 1* and *2*). The trypanosome flagellum emerges into an enclosed structure called the flagellar pocket, which is the site of all exocytosis and endocytosis (*Overath and Engstler, 2004*). CFB2-depleted cells often had grossly enlarged flagellar pockets. This defect was previously observed after depletion of clathrin or actin, when it is caused by a failure in endocytosis (*Allen et al., 2003*; *García-Salcedo et al., 2004*), and also in cells with flagellar assembly defects (*Broadhead et al., 2006*). Various abnormal vesicular structures were also seen, consistent with the previous report (*Ooi et al., 2018*; *Figure 6*, *Figure 6—figure supplements 1* and *2*).

## The F-box is implicated in auto-regulation and MKT interaction is required for mRNA activation

Results of a high-throughput study suggested that the half-life of untagged CFB2 is probably less than 1 hr (*Tinti et al., 2019*), so we wondered whether the interaction of CFB2 with SKP1 provokes its own degradation. An affinity-purified antibody faintly recognized a ~50 KDa protein in Western blots of trypanosome lysates, probably with less than 1000 molecules per cell (*Figure 7—figure supplement 1*). The abundance of this protein was increased by prior incubation of the cells with the proteasome inhibitor MG132 (*Figure 7—figure supplement 1*), suggesting that it is unstable. Tagging of both the N- and C-terminus did not increase the abundance (*Figure 7—figure supplement 1*) or the effect of MG132 (*Figure 7—figure supplement 1*). We therefore suspected that the F-box interaction with SKP1 might be causing CFB2 instability.

Results from tethering screens had suggested that the C-terminal portion of CFB2 is able to activate expression of a reporter mRNA but the full-length protein does not (*Lueong et al., 2016*; *Erben et al., 2014*). All fragments with activation function contained the C-terminal MKT1 interaction domain, YRHDPY, and lacked the F-box (*Figure 7—figure supplement 2*); but many also included a conserved region with basic and hydrophobic residues (illustrated in *Figure 7—figure supplement 2*). To examine the functions of both the SKP1 and MKT1 interaction motifs, we expressed various mutants as fusion proteins, with the lambdaN peptide at the N-terminus and a myc tag at the C-terminus, testing their expression and ability to enhance expression of a boxB reporter. Both deletion and mutation of the F-box motif increased fusion protein abundance (*Figure 7B*, *Figure 7—figure supplements 1* and *3*), implicating the F-box in auto-regulation of the stability ectopically expressed CFB2 protein. Increased abundance may explain why the full-length F-box mutant was able to activate reporter expression better than the wild-type protein, although not as much as the C-terminal fragment (*Figure 7C*, *Figure 7—figure supplement 3*). As expected, an intact MKT1 interaction motif was essential for activation. The lack of activation by the full-length protein could at least in part be caused by recruitment of SKP1 to the tethered, RNA-bound protein: we do not know whether this activity is prevented when CFB2 is bound via its own RNA-binding domain.

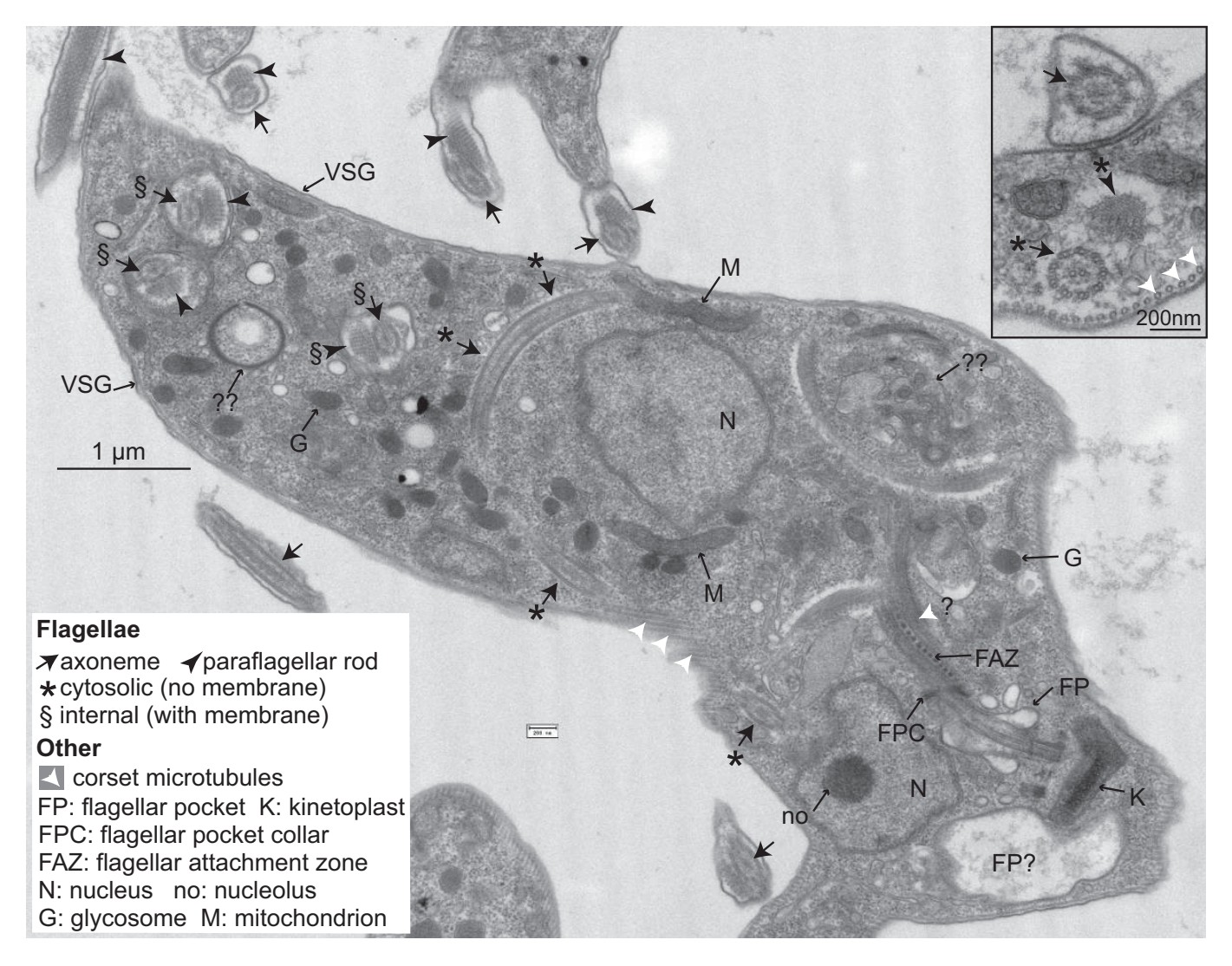

**Figure 6.** Trypanosomes lacking CFB2 (cyclin F-box protein 2) have internal flagella. A typical trypanosome 16 hr after induction of RNAi targeting *CFB2*. The central cell has one flagellar pocket (FP) of normal appearance; the base of the emerging flagellum can be seen. This should exit the cell at the flagellar pocket collar (FPC) and may then run beneath the indicated flagellar attachment zone (FAZ). Next to the neighbouring kinetoplast (K), which has normal morphology, there is a second vacuole that contains low-density material; this could be part of a second, much enlarged, flagellar pocket (the usual FP diameter does not exceed 500 nm; *Lacomble et al., 2009*). The cell has two nuclei (N), and in one the section passes through the nucleolus (no). The cell contains at least three additional flagellar axonemes. One (*) is found in both longitudinal and transverse cross-sections and lacks a surrounding membrane. Additional axonemes with paraflagellar rods (§) are surrounded by double membranes that may lack the thick variant surface glycoprotein (VSG) coat (indicated at two positions on the cell exterior). The paraflagellar rod is normally seen only on flagella that have exited the FP. The glycosomes (G) and mitochondrion (M) appear normal. Double question marks indicate abnormal membrane structures that are suggestive of autophagy and internal membrane proliferation. The inset shows part of an enlarged flagellar pocket with a membrane-enclosed axoneme; beneath it are the four specialized microtubules that are found next to the FAZ; these usually, however, are associated with endoplasmic reticulum. The section also includes a membrane-less internal flagellum and flagellar rod. More images are in *Figure 6—figure supplements 1* and *2*.

The online version of this article includes the following figure supplement(s) for figure 6:

**Figure supplement 1.** Effects of CFB2 (cyclin F-box protein 2) RNAi on ultrastructure – additional images.

**Figure supplement 2.** Effects of CFB2 (cyclin F-box protein 2) RNAi on ultrastructure – additional images.

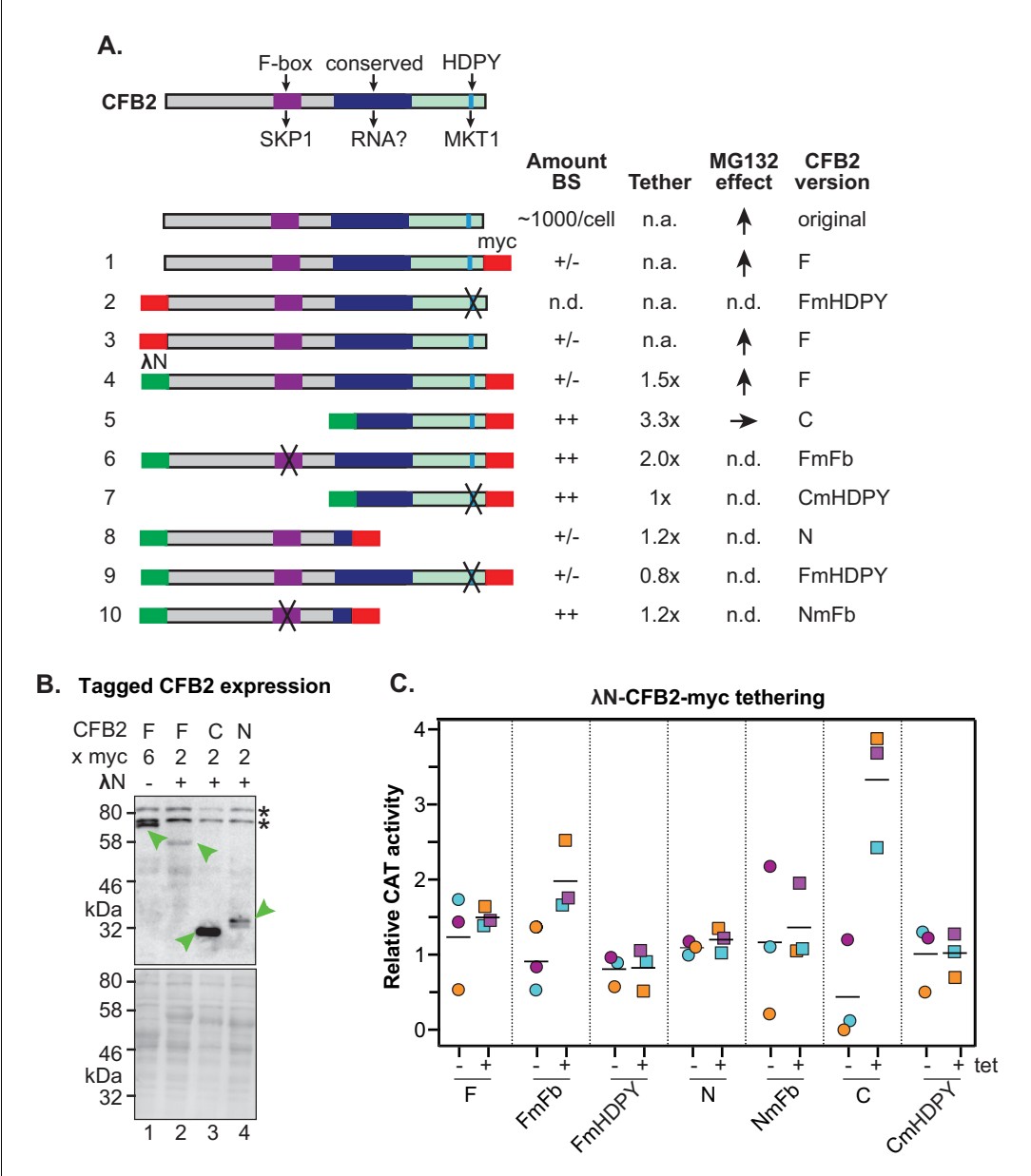

**Figure 7.** Roles of the SKP1 and MKT1 interaction domains of CFB2 (cyclin F-box protein 2). (A) Structures of different expressed versions of CFB2, with expression level, effects when tethered, and effect of MG132 on the expression level as shown in panels **B and C**. For expression, +/- means difficult to detect, and ++ is very easy to detect using anti-myc antisera, with double myc tags at either N- or C-terminus as indicated. Constructs 1 and 3 exist in two versions. Versions with an rRNA promoter and 6xmyc tags were used for measurement of the MG132 effect in bloodstream forms. Versions with an inducible promoter and two myc tags were used to assess the effect of CFB2 expression in procyclic forms. All other constructs have two myc tags and an inducible promoter. Versions 4–10 have the N-terminal lambdaN peptide. 'F' means 'full length', 'C' means C-terminal portion, 'N' means N-terminal portion. Mutations in the F-box (Fb) and in the HDPY motif are illustrated in *Figure 3*. Abbreviations: n.a., not applicable; n.d., not determined. (B) Expression of fusion proteins in bloodstream forms. Lane 1 shows constitutive expression of CFB2-6xmyc (construct 1) from an rRNA promoter, while lanes 2–4 show expression of different lambdaN-2xmyc-tagged fragments (*Lueong et al., 2016*; *Zoltner et al., 2018*; *Günzl et al., 2003*) after 24 hr induction of expression from a tetracycline-inducible EP procyclin promoter. The myc epitope was detected (green arrow). Whole-cell proteins were visualized by Ponceau (bottom). (C) Graph showing effects of tethering different versions of CFB2 (constructs 4–10). The bloodstream-form trypanosomes used expressed a *CAT* mRNA followed by five boxB loops, then the actin 3'-untranslated region (3'-UTR). Three independent cell lines were selected for each CFB2 plasmid and CAT activity was measured with or without a 24 hr incubation with 100 ng/ml tetracycline to induce fusion protein expression (*Figure 7—figure supplement 1*).

The online version of this article includes the following figure supplement(s) for figure 7:

**Figure supplement 1.** Antibody tests and CFB2 (cyclin F-box protein 2) protein expression.

*Figure 7 continued on next page*

*Figure 7 continued*

**Figure supplement 2.** Tethering screen results and the conserved region C-terminal to the F-box.
**Figure supplement 3.** Expression of CFB2 (cyclin F-box protein 2) lambdaN fusion proteins in bloodstream forms.

## The action of CFB2 depends on a conserved 16-mer in the 3'-UTR of the *VSG* mRNA

*VSG* mRNA 3'-UTRs are relatively short, consisting of a rather variable CU-rich domain followed by 8-mer and 16-mer sequences that are conserved in almost all available *VSG* cDNA sequences (***Figure 8—figure supplement 1***). The 16-mer sequence in particular is highly specific to *VSG* transcripts, whereas CU-rich domains are present in numerous other 3'-UTRs. Previous experiments using reporter mRNAs have therefore focused on the 8-mer and the 16-mer. These showed that the 16-mer is required for high *VSG* mRNA abundance and stability in bloodstream forms (***Ridewood et al., 2017***; ***Berberof et al., 1995***). It is also required for $m^6A$ modification of the poly(A) tail, which plays a role in *VSG* mRNA stabilization (***Viegas et al., 2020***). To find out which sequences are responsive to CFB2, we used a construct in which *GFP* reporter mRNAs with various 3'-UTRs (***Figure 8A***) are expressed from an rRNA promoter, which results in constitutive RNA polymerase I transcription. The *trans* splicing signal and 5'-UTR of the reporter are derived from the *EP1* procyclin locus. The polyadenylation site, and polyadenylation efficiency, are wholly dictated by the downstream intergenic region and splicing signal, which are derived from the actin (*ACT*) locus and are also responsible for processing the mRNA encoding the selectable marker. Thus, all changes in GFP reporter expression are caused by differences in mRNA translation and decay. Since our studies so far had concentrated only on mRNAs with the *VSG2* 3'-UTR, we now instead made reporters with the 3'-UTR from *VSG4*. The two UTRs share the 8-mer and 16-mer, but the preceding and intervening regions show differences in both sequence and length (***Figure 8—figure supplement 1***). As expected from previously published results (***Ridewood et al., 2017***), the presence of the 16-mer resulted in relatively high expression in bloodstream forms (*GFP-VSG*); its mutation to a scrambled version (*GFP-VSGm16*) decreased the GFP protein and mRNA levels to approximately the same level as the actin (*GFP-ACT*) control, while scrambling of the 8-mer (*GFP-VSGm8*) had no effect (***Figure 8B***, ***Figure 8—figure supplements 2*** and ***3***).

In procyclic forms, CFB2 is not expressed. Upon transformation of the cloned trypanosomes to the procyclic form, the only *VSG* reporter giving detectable expression was that in which the 8-mer had been mutated (*GFP-VSGm8*) (***Figure 8B***, ***Figure 8—figure supplements 2*** and ***3***). This result implicates the 8-mer in ensuring the low abundance of any *VSG* mRNAs that are made in the procyclic form.

We next inducibly expressed C-terminally myc-tagged CFB2 (CFB2-myc) in procyclic forms expressing the different reporters. As a control we expressed CFB2-myc with a mutated MKT1 interaction domain. These proteins were easier to detect than in bloodstream forms, suggesting that the auto-regulation has a degree of stage specificity (***Figure 8—figure supplement 4***). Expression of CFB2-myc had no effect on GFP expression if the actin 3'-UTR was used (*GFP-ACT*), but caused marked increases in expression with the *VSG* 3'-UTR was used (*GFP-VSG*) (***Figure 8C***, ***Figure 8—figure supplement 4***). Expression of the reporter with the 8-mer mutation (*GFP-VSGm8*) was also enhanced (***Figure 8C***, ***Figure 8—figure supplement 4***). Similarly, preliminary results derived from a single clone indicated that expression from a reporter with a scrambled upstream region (GFP-VSGups) was also enhanced upon CFB2 expression (***Figure 8—figure supplement 4***).

In contrast, the reporter with mutant 16-mer was unaffected by CFB2-myc expression (*GFP-VSGm16*) (***Figure 8C***, ***Figure 8—figure supplement 4***). As expected, the HDPY mutant of CFB2, which cannot interact with MKT1, had no effect on reporter expression. In some clones, the HDPY mutant was also equally expressed in the presence and absence of tetracycline, indicating a lack of selection for strong regulation, whereas the wild-type protein was undetectable without induction (***Figure 8—figure supplement 4***).

To confirm that the effect of CFB2 on the reporters was indeed caused by specific binding, we immunoprecipitated CFB2-myc and detected the reporter mRNAs by reverse transcription and RT-qPCR. We then compared the amounts of precipitated *GFP* mRNA to the background precipitation of *TUB* mRNA. Expression of *GFP-VSGm16* was unfortunately too low for quantitative detection.

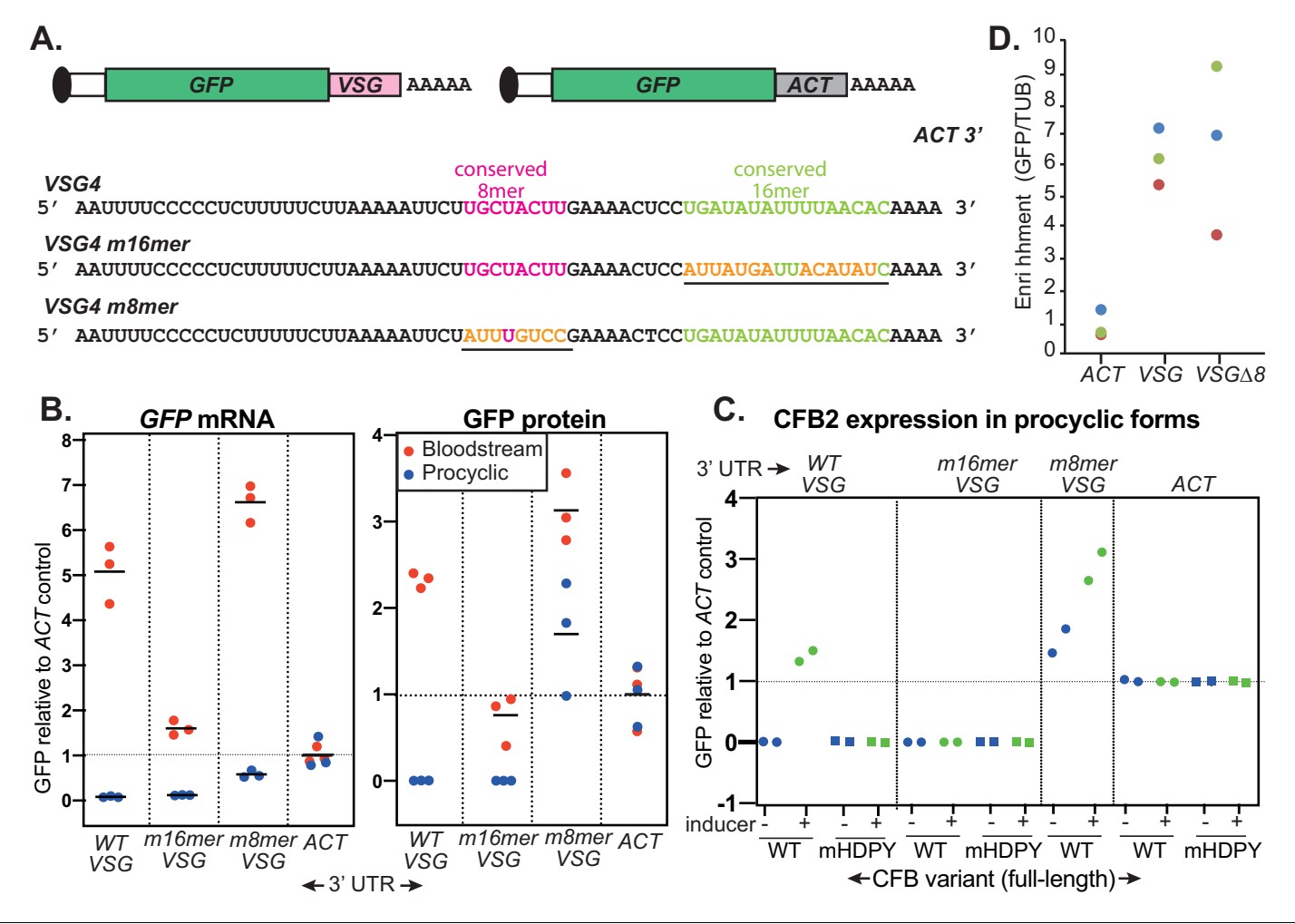

**Figure 8.** The role of the variant surface glycoprotein (*VSG*) 3'-untranslated region (3'-UTR). (**A**) Diagram of alternative reporters and sequences of the 3'-UTRs. The 3'-UTR from the *VSG4* gene was cloned downstream of *GFP*, with polyadenylation directed by the *trans* splicing signal of the puromycin resistance cassette. After transfection of the linearized plasmid into trypanosomes, it integrates into an rRNA spacer and is transcribed from an rRNA promoter. (**B**) EATRO1125 *Trypanosoma brucei* were transfected with the reporters as bloodstream forms (*red*), then three independent cloned lines for each construct were transformed to procyclic forms (*blue*). Levels of GFP were quantified by Western blotting (***Figure 8—figure supplement 2***), and of mRNA by Northern blotting (***Figure 8—figure supplement 3***). All mRNAs were of the expected size. Results were normalized to the average for three controls expressing *GFP* mRNA with an actin (*ACT*) 3'-UTR. The left-hand panel shows *GFP* mRNA expression, and the right-hand panel, GFP protein. Individual measurements are shown; the bar is the arithmetic mean. (**C**) Effect of CFB2 (cyclin F-box protein 2)-2xmyc expression on GFP expression in procyclic forms. For each GFP reporter from (**B**), we selected the procyclic clone with median expression. This was transfected with CFB2-inducible expression constructs 2 (FmHDPY mutant; *green*) and 3 (wt; *blue*) from ***Figure 7A*** and two independent cell lines were selected for each CFB2 plasmid. GFP protein was measured after 24 hr incubation with 100 ng/ml tetracycline and quantified by Western Blot (***Figure 8—figure supplement 4***), and results were normalized to the average for four controls expressing *GFP* mRNA with an actin (*ACT*) 3'-UTR. (**D**) Tagged CFB2 binds to *GFP* mRNAs via the *VSG* 3'-UTR. CFB2-2xmyc was immunoprecipitated from the cells shown in (**C**), and RNA was prepared. The relative amounts of *GFP* and tubulin (*TUB*) RNA were then measured by qPCR and the amount in the bound, relative to the unbound, fraction was calculated. Results were then normalized by dividing the result for *GFP* by the result for *TUB*. On the graph, results for three biological replicates are shown in different colours.

The online version of this article includes the following figure supplement(s) for figure 8:

**Figure supplement 1.** Alignment of variant surface glycoprotein (*VSG*) 3'-untranslated regions (3'-UTRs), downloaded from GenBank.

**Figure supplement 2.** Western blots used for ***Figure 8***.

**Figure supplement 3.** Northern blots used for ***Figure 8***.

**Figure supplement 4.** Expression of different versions of CFB2 (cyclin F-box protein 2)-myc in cells with GFP reporters.

However, compared with *GFP-ACT*, *GFP-VSG* mRNA was sixfold more abundant in the CFB2 pull-downs, and similar enrichment was seen for *GFP-VSGm8*.

The results confirmed that CFB2 binds specifically to the *VSG* 3'-UTR and that binding is unaffected by mutation of the conserved 8-mer. Most importantly, they showed that CFB2 is able to increase expression from an mRNA bearing the *VSG* 3'-UTR and that this depends on the conserved 16-mer sequence.

## Discussion

In this paper we describe a robust procedure to purify mRNAs and their associated proteins. Success of such procedures depends on their use to target mRNAs that have relatively high abundance; since we could identify proteins that were specifically associated with the alpha-TUB mRNA, the threshold probably lies somewhere below 1% of total mRNA. The major limitation is the fact that mRNAs are about a thousand times less abundant than protein in cell lysates, so that a one-step purification is bound to be dominated by abundant protein contaminants. Because of this, most previous successful attempts at mRNA-centric purification of interacting proteins have employed in vitro affinity purification using relatively short RNAs as baits (reviewed in *Gerber, 2021*). Purification of proteins bound in vivo to abundant structural RNAs is, in contrast, less challenging (e.g. *McHugh et al., 2015*). To our knowledge, there has so far been only one other report of successful characterization of specific native mRNPs. The approach used was conceptually very similar to ours, although the reported results relied on GFP reporters, which potentially loses coding region-specific information (*Theil et al., 2019*). In contrast, we show here that when native sequences are purified, our procedure can be used to purify at least two – and probably, several more – different mRNPs from the same extracts. This is useful not only in saving biological material, but also because, if the order of RNA selection is varied, the different purifications act as internal specificity controls.

The role of the 3'-UTR in *VSG* mRNA abundance was demonstrated a quarter of a century ago (*Berberof et al., 1995*), and conservation of 3'-UTR sequences was reported even earlier (*Majumder et al., 1981*). It was also known that the action of the 3'-UTR is stage specific (*Berberof et al., 1995*), and that *VSG* mRNA is degraded during differentiation to the procyclic form (*Overath et al., 1983*). By purifying the *VSG* mRNA together with its associated proteins, we have now identified CFB2 as the protein responsible for *VSG* mRNA retention in bloodstream forms. Moreover, we could demonstrate the mechanism by which CFB2 acts: recruitment of a stabilizing complex that includes MKT1, PBP1, PABP2, and the cap-binding translation initiation complex EIF4E6/G5. CFB2 action depends on the presence of a conserved 16-mer in the *VSG* mRNA 3'-UTR. This sequence is also required for m6A modification of the *VSG* poly(A) tail (*Viegas et al., 2020*). In principle, CFB2 might recognize either the 16-mer or m6A. However, m6A is present on many mRNAs in addition to *VSG* (*Viegas et al., 2020*), whereas CFB2 depletion resulted in highly specific *VSG* loss. We therefore suggest that CFB2 recognizes the 16-mer.

Although *VSG* transcription is shut off in procyclic forms, the parasites appear to have a fail-safe mechanism that ensures that any accidentally produced *VSG* mRNA is rapidly degraded. We find here that a conserved 8-mer sequence in the *VSG* 3'-UTR is implicated in this. Loss of *VSG* mRNA stability during differentiation is therefore the result of a combination of two processes: the disappearance of CFB2, and the expression of another, as yet unknown, protein that binds to the 8-mer, represses translation and causes mRNA destruction.

Humans have at least 15 proteins that contain both RNA-binding and E3 ligase domains (*Cano et al., 2010*); they are implicated in auto-ubiquitination, and in either mono-ubiquitination or polyubiquitination of other targets (*Cano et al., 2015*; *Liu et al., 2016*; *Zhang et al., 2015*; *Athanasopoulos et al., 2016*; *Td and Wilkinson, 2018*; *Feng et al., 2017*). Our results suggest that the interaction of CFB2 with SKP1 promotes CFB2 degradation, presumably through auto-ubiquitination. This may fine-tune CFB2 abundance to limit VSG synthesis to an ideal level, preventing secretory pathway overload. SKP1 was detectable in purified MKT1-containing complexes (*Singh et al., 2014*; *Melo do Nascimento et al., 2020*), so CFB2 may be able to interact with SKP1 and MKT1 simultaneously. We do not know whether CFB2 has additional ubiquitination targets. These interactions, as well as the mode of CFB2 mRNA binding, require further investigation.

## Materials and methods

### *T. brucei* growth and manipulation

Bloodstream-form Lister427 or EATRO1125 *T. brucei* expressing the *tet* repressor (pHD1313 or 2T1 cells) (*Alibu et al., 2005*; *Alsford and Horn, 2008*) were grown in HMI-11 medium (PAN Biotech). Cells with selectable VSG2 expression were a kind gift from Gloria Rudenko (Imperial College). These are 'single marker' derivative cell line carrying a puromycin resistance gene incorporated immediately behind the promoter of the active VSG2 ES (SM221) (*Narayanan et al., 2011*). Puromycin, phleomycin, hygromycin, G418, and blasticidin were used at 0.2, 0.2, 5, 2, and 5 µg/ml, respectively, for selection of recombinant clones. Regulation by the *VSG* 3'-UTR was investigated using EATRO1125 bloodstream-form trypanosomes, which were converted to procyclic forms by incubation with cis-aconitate and transfer to 27°C as previously described (*Mugo and Clayton, 2017*).

### Plasmids

Some cloning reactions were carried out with NEBuilder HiFi DNA assembly cloning kit (NEB, #E5520) while others were made by conventional means. The plasmids are listed in *Supplementary file 1*. For RNAi of *CFB2*, a specific attB-tagged gene fragment (Tb927.1.4650, 378 bp) was amplified and cloned into pGL2084 (*Jones et al., 2014*) by Gateway recombination. The streptavidin-binding protein (SBP) tagged lambda-GFP construct was generated by amplifying the lambda-GFP and SBP coding regions in pHD2294 and pcDNA4-MS2-CP-GFP-SBP (a gift from J. Gerst – Weizmann Institute), respectively, into *Hind* III and *Bam* HI sites of the pRpa vector (*Alsford et al., 2005*). For over-expression of CFB2, we cloned the full ORF into the pRpa plasmid to add 6 C-terminal myc tags. We then subcloned the CFB2-6xmyc ORF into pHD1991, which drives constitutive expression and is inserted into the RRNA locus (*Delhi et al., 2011*). For aptamer tagging at the VSG2 native locus, a 225 bp fragment containing five *boxB* repeats in pHD2277 (*Erben et al., 2014*) was amplified and cloned into a derivative pSY37F1D-CTR-BSD plasmid (*Pinger et al., 2017*) containing the *VSG2* CDS and 3'-UTR sequence to generate a *VSG2* gene with a boxB immediately after its endogenous 3'-UTR. Before transfection, the plasmid was linearized with *Bgl* II. Regulation by the *VSG* 3'-UTR was investigated using plasmids containing the *GFP* coding region and a puromycin selection cassette.

Prior to transfection into trypanosomes, the RNAi and CFB2 C-terminal myc-tag over-expression cassettes were digested with *Asc* I; all plasmids with pHD numbers were linearized with *Not* I. Linearized constructs were transfected into 2T1 cells (*Alsford and Horn, 2008*) or cells containing integrated pHD1313 (*Alibu et al., 2005*).

### RNA antisense purification mass spectrometry

To identify proteins specifically interacting with *VSG2* and alpha-*TUB* mRNAs, we UV-cross-linked RNA and proteins in vivo and captured RNA using biotinylated oligonucleotides using a protocol modified from *McHugh and Guttman, 2018*. To reduce the costs associated with growth media, we captured the two target RNAs in successive cycles from the same sample, exchanging the order of the probes to equalize the chance for contaminants. Mass spectrometry was done on pooled samples from four technical replicates (two each for each order), and the entire procedure was performed three times.

#### UV cross-linking

Cells at mid-log phase were collected (1100 × $g$ for 5 min), resuspended into vPBS (PBS supplemented with 10 mM glucose and 46 mM sucrose, pH 7.6), then UV cross-linked on ice using two rounds of 0.18 J/cm$^2$ of UV at 254 nm (Analytik Jena). Cells were collected, washed once with PBS, and pellets were flash-frozen in liquid nitrogen for storage at −80°C.

#### Total cell lysate preparation

We lysed batches of ~9×10$^9$ cells by completely resuspending frozen cell pellets in 25 ml ice-cold urea-based cell hybridization buffer (25 mM Tris pH 7.5, 500 mM LiCl, 0.25% dodecyl maltoside, 0.2% sodium dodecyl sulphate, 0.1% sodium deoxycholate, EDTA 5 mM, TCEP 2.5 mM, and 4 M

urea). Next, the cell sample was passed ~10 times through a 27-gauge needle attached to a 25 ml syringe in order to disrupt the pellet and shear genomic DNA. At this point lysates were kept at −80°C or incubated at 65°C for 10 min before clearing by centrifugation for 10 min at 10,000 × *g*.

For the first purification, frozen cell pellets were resuspended in 8 ml ice-cold detergent-based cell lysis buffer (10 mM Tris pH 7.5, 500 mM LiCl, 0.5% dodecyl maltoside, 0.2% sodium dodecyl sulphate, 0.1% sodium deoxycholate, 1× Protease Inhibitor Cocktail EDTA-free, and 900 U of Murine RNase Inhibitor) (New England Biolabs), and cell sample passed 10–15 times through a 27-gauge needle attached to a 25 ml syringe in order to disrupt the pellet and shear genomic DNA. The samples were then treated for 10 min at 37°C adding 1× DNAse salt solution (2.5 mM MgCl$_2$, 0.5 mM CaCl$_2$), and 900 U of DNase I [Roche] to digest DNA. Samples were returned to ice and the reaction was immediately terminated by the addition of 16 ml 1.5× cold hybridization buffer to stop reaction.

## RNA antisense purification of cross-linked complexes

One millilitre of hydrophilic streptavidin magnetic beads (New England Biolabs) were washed five times with equal volume of hybridization buffer. Lysate samples were pre-cleared by incubation with the washed magnetic beads at 37°C for 30 min with intermittent shaking at 1100 rpm on an Eppendorf Thermomixer C (30 s mixing, 30 s off). Streptavidin beads were then magnetically separated from lysate samples using a Dynal magnet (Invitrogen). The beads used for pre-clearing lysate were discarded and the lysate sample was transferred to fresh tubes twice to remove all traces of magnetic beads.

For each mRNA, we used seventeen 90-mer 5′-biotinylated complementary DNA oligonucleotides that spanned the entire length of the target RNAs. The oligonucleotides were heat-denatured at 85°C for 3 min and then snap-cooled on ice. Probes and pre-cleared lysate were mixed and incubated at 65°C using an Eppendorf Thermomixer with intermittent shaking (30 s shaking, 30 s off) for 1.5 hr to hybridize probes to the target RNA. Samples were then incubated with washed C1 streptavidin-coated magnetic beads (Thermo Fisher Scientific) at 65°C for 2.5 hr on an Eppendorf Thermomixer C with intermittent shaking as above. Beads with captured hybrids were washed five times with hybridization buffer at 65°C for 5 min to remove non-specifically associated proteins. Then, samples were washed twice with DNAse buffer (50 mM Tris pH 7.5, 300 mM LiCl, 0.5% NP40, 0.1% NLS, and 1× DNAse salt solution) and incubated with intermittent shaking (30 s shaking, 30 s off) for 15 min with 15 units of DNAse I (Roche) to remove DNA traces. Around 2% of the total beads were removed and transferred to a fresh tube after the final wash to test RNA capture by RT-qPCR. The remaining beads were resuspended in Benzonase Elution Buffer (20 mM Tris pH 8.0, 2 mM MgCl$_2$, 0.05% NLS, 0.5 mM TCEP) for subsequent processing of the protein samples.

## Elution of protein from the beads

Elution of captured proteins from streptavidin beads was achieved by digesting all nucleic acids with 125 U of Benzonase nonspecific RNA/DNA nuclease for 2 hr at 37°C. Beads were then magnetically separated from the sample using a Dynal magnet (Invitrogen); the supernatant containing eluted proteins was first partially concentrated in a Speedvac to about 200 µl and finally methanol/chloroform precipitated (*Wessel and Flügge, 1984*).

## Elution and analysis of RNA samples

Beads with hybrids were magnetically separated using a Dynal magnet and the supernatant was discarded. Beads were then resuspended by pipetting in 50 µl NLS RNA Elution Buffer (20 mM Tris pH 8.0, 10 mM EDTA, 2% NLS, 2.5 mM TCEP). To release the target RNA, beads were heated for 2 min at 95°C. Beads were then magnetically separated and the supernatants containing eluted target RNA were digested by the addition of 1 mg/ml proteinase K for 1 hr at 55°C to remove all proteins. The remaining nucleic acids were purified using the RNA Clean and Concentrator Kit (Zymo).

For synthesis of cDNA, the Maxima First Strand cDNA Synthesis Kit for RT-qPCR (Thermo Fisher Scientific) was used following manufacturer's protocol. To quantify RNA enrichment, RT-PCR was performed in triplicate using Luna Universal qPCR Master Mix (New England Biolabs) with variable amounts of cDNA and 0.5 µM of target-specific primers in a CFX connect instrument (Bio-Rad). Primer sequences are shown in *Supplementary file 1*.

## Mass spectrometry analysis

Proteins from four independent purifications (material from about of $3 \times 10^{10}$ trypanosomes, two with TUB mRNA selected first and two with *VSG*) were pooled, subjected to denaturing SDS-PAGE, and analysed by the ZMBH Core Facility for mass spectrometry and proteomics as previously described (*Lueong et al., 2016*). Data were quantitatively analysed using Perseus (*Tyanova et al., 2016*).

## RNA-binding protein purification and identification

Rapid experiments and controls: To identify CFB2 specifically interacting with VSG2 mRNA, we performed captures of lambda-GFP-SBP proteins in cells expressing boxB-tagged *VSG2* mRNA and lambdaN-GFP-SBP. Cells lacking either component were used as controls.

## Formaldehyde cross-linking and cell lysis

Cells at mid-log phase were collected, resuspended in PBS, and then cross-linked with 0.01% formaldehyde at room temperature for 10 min with slow shaking. The cross-linking reaction was terminated by adding 1 M glycine-NaOH buffer (pH 8.0) to a final concentration of 0.125 M and additional shaking for 2 min. The cells were then washed once with ice-cold PBS buffer and the pellet was flash-frozen in liquid nitrogen, and stored at −80°C. Cell pellets were thawed by addition of ice-cold lysis buffer (25 mM Tris-HCl, pH 7.5, 175 mM KCl, 0.5% NP40, 1 mM DTT, and 120 U RNAse inhibitor) (New England Biolabs) supplemented with EDTA-free protease inhibitors (Roche). Samples were then subjected to three cycles of sonication (10 pulses of 0.5 s) followed by 1 min rest between cycles at 4°C (Branson Ultrasonics Sonifier S-250). The extract was then supplemented with 1x DNAse salt solution and 100 U DNAse I (Roche) and incubated for 30 min at 4°C on rotator. Samples were returned to ice and the reaction was immediately terminated by the addition of 10 mM EDTA and 10 mM EGTA. The extract was finally clarified by centrifugation for 20 min at $15,000 \times g$, the supernatant removed to a new microcentrifuge tube and protein concentration determined using the Bradford assay (Bio-Rad).

## Precipitation of RNP complexes

RNA-binding protein purification and identification (RaPID) was performed essentially as described by *Slobodin and Gerst, 2010* with a few modifications. In order to block endogenous biotinylated moieties, the protein aliquot taken for pull-down was incubated with 10 mg of free avidin (Sigma) per 1 mg of protein input at 4°C for 1 hr with constant rotation. In parallel, streptavidin magnetic beads (New England Biolabs) were washed five times with 1 ml of lysis buffer supplemented with EDTA 5 mM. Pull-down was then performed by adding the indicated amount (see figure legends) of avidin-blocked total cell lysate to the beads, followed by incubation at 4°C for 3 hr with constant rotation. Beads with captured mRNPs were washed three times with lysis buffer and three times with washing buffer (25 mM Tris-HCl, pH 7.5, 300 mM KCl, 0.5% NP40, 1 mM DTT, 5 mM EDTA, and 40 U/ml RNAse inhibitor) (New England Biolabs) at 4°C for 5 min to remove non-specifically associated proteins. For elution of the cross-linked RNP complexes, 250 μl of washing buffer supplemented with 6 mM free biotin (Sigma) was added to the beads, followed by 1 hr of incubation at 4°C with rotation. To reverse the cross-link for RNA analysis, samples were incubated at 70°C for 1 hr with cross-link reversal buffer (50 mM Tris-HCl, pH 7.5, 5 mM EDTA, 10 mM DTT, and 1% SDS).

## Protein blotting and antibodies

Protein samples were run according to standard protein separation procedures, using SDS-PAGE. The following primary antibodies were used: mouse α-GFP (1:2000; Roche Cat# 11814460001, RRID: AB_390913), mouse α-myc (1:2000; Santa Cruz Biotechnology Cat# sc-40, RRID:AB_627268), rat α-RBP10 (1/500; RRID:AB_2890154), rabbit α-aldolase (1/2000; RRID:AB_2890155). We used horseradish peroxidase coupled secondary antibodies (1:2000; Bio-Rad Cat# 170–6516, RRID:AB_11125547; Bio-Rad Cat# 170–6515, RRID:AB_11125142). Blots were developed using an enhanced chemiluminescence kit (Amersham; RPN2209) according to the manufacturer's instructions. Densitometry was performed using Fiji v. 2.0.0.

## Immunoprecipitation of mRNA-protein complexes

For immunoprecipitation, cells expressing CFB2-6xmyc were first cross-linked using either UV irradiation or formaldehyde as described above. In some cases, MG132 (Cayman; Item No. 10012628) was added to a final concentration of 10 µg/ml for 1 hr before cross-linking.

## Formaldehyde-treated cell extracts were made as described for RaPiD

UV irradiated cells were washed in cold PBS and the cell pellet snap-frozen in liquid nitrogen or used immediately. Cells were lysed by resuspension in lysis buffer (50 mM Tris-HCl, pH 7.5, 2 mM MgCl$_2$, 10 mM KCl, 0.1 mM DTT, 0.5% (w/v) NP-40, 100 U/ml murine RNase inhibitor [NEB], and EDTA-free protease inhibitor cocktail [Roche]). Next, samples were passed ~10 times through a 27-gauge needle in order to disrupt the pellet and shear genomic DNA. After adding KCl to 150 mM, the lysate was centrifuged at 4°C for 10 min at 12,000 $\times$ *g*.

The supernatants from both types of extract were subjected to immunoprecipitation using anti-c-Myc magnetic beads (Pierce) for 3 hr at 4°C. After extensive washing in lysis buffer supplemented with the indicated amount of KCl (see figure legends), RNA was extracted from the immunoprecipitated material using Trifast reagent (Peqlab, GMBH).

## RNAi by direct dsRNA transfection

The template for *CFB2* dsRNA production was identical to the one employed in the tet-inducible system and was amplified using primers that include a T7 promoter for in vitro transcription (*Supplementary file 1*). dsRNA was made using the MEGAscript RNAi Kit (Ambion, Thermo Fisher Scientific). 20 µg of dsRNA or the same buffer volume (for the mock control) were transfected into 30 million cells, using the X-001 program of an AMAXA Nucleofector (Lonza).

## Electron microscopy

For transmission electron microscopy, samples were prepared exactly as described in *Höög et al., 2010* after 16 hr of CFB2 RNAi induction. The blocks were sectioned using a Leica UC6 ultramicrotome (Leica Microsystems Vienna) in 70 nm thin sections. The sections were placed on formvar-coated grids, post-stained and imaged on a JEOL JEM-1400 electron microscope (JEOL, Tokyo) operating at 80 kV and equipped with a 4K TemCam F416 (Tietz Video and Image Processing Systems GmBH, Gautig).

## RNASeq

RNA was prepared after 9 hr of *CFB2* RNAi induction in cells expressing *VSG2*. rRNA was depleted using complementary oligonucleotides and RNaseH as previously described (*Minia et al., 2016*). It was fragmented, and cDNA libraries were prepared using an Illumina kit before sequencing (MiSeq). Raw reads trimmed then aligned to *T. brucei* genomes using TrypRNASeq (*Leiss et al., 2016*) and genomes downloaded from TritrypDB. To assess levels of transcripts from expression sites, reads were aligned to the 2018 version of the Lister427 genome (*Müller et al., 2018*). Each read was allowed to align once (such that reads for repeated genes are distributed among the copies) and these results are in *Supplementary file 3*. To obtain more information about annotation of any affected chromosome-internal genes, we repeated the alignment using the well-annotated TREU927 genome (*Berriman, 2005*; *Supplementary file 4*). To look for changes in expression, we used DeSeqU1 (*Leiss and Clayton, 2016*), a user-friendly RStudio DeSeq2 (*Love et al., 2014*) application. To look for enrichment of particular functional classes, we considered a list of genes in which each individual sequence is present only once (i.e. additional gene copies have been removed) (*Siegel et al., 2010*). Since, in this list, reads from repeated genes will be under-represented, we multiplied each read count by the gene copy number. Copy numbers were obtained by repeating the alignments, allowing each read to align 20 times.

The distribution of reads across Lister427 chromosomes was visualized using Artemis. For this, we chose induced and uninduced samples with similar total aligned read counts.

## CFB2 expression in *Escherichia coli*

For expression of CFB2 in *E. coli*, the coding sequence of CFB2 was cloned into pET-NusA for N-terminal tagging with a 6×His-tag and the transcription termination/antitermination protein NusA,

which was then transformed into BL21 competent *E.coli*. Individual transformants were inoculated in 5 ml of LB$^{Kan}$ each and grown for 16 hr at 27°C. After that, OD$_{600}$ measurements were performed every hour, until the cultures reached an OD$_{600}$ of 0.8. For each inoculum, 1 ml was collected by centrifugation (2 min, 16,000 × *g*), and the supernatant was subsequently removed. The pellet was then lysed in Laemmli buffer ('uninduced' sample) and stored at −20°C until further use. Recombinant protein expression was induced by addition of IPTG at a final concentration of 1 mM to the remaining 4 ml of the culture. Furthermore, another 1 ml of LB$^{Kan}$ supplemented with IPTG (1 mM) was added, after which the samples were incubated at 27°C for 14 hr. Upon reaching an OD$_{600}$ of 2.5–3, 300 µl were collected per sample by centrifugation at 16,000 × *g* for 2 min. These 'induced' samples were processed and lysed similar to the 'uninduced' samples. For analysing protein expression, the samples were boiled at 95°C for 10 min and subsequently separated using 10% SDS-polyacrylamide gels, which were then stained by Coomassie staining.

CFB2 was also expressed as a fusion with 6 His tags and the GB protein, by cloning into pETGB. Expression was induced and inclusion bodies purified as above. This protein was used for rabbit-antisera purification and as a positive control on Western blots.

## Analysis of soluble and insoluble *E. coli* fractions

Induction and harvest of the cultures were performed as described above. However, instead of lysing the cells with Laemmli buffer, the pellet was resuspended in 350 µl of bacterial lysis buffer (50 mM Tris [pH 7.5], 100 mM NaCl, 1 mM EDTA, 1 mM DTT, 1 µg/ml aprotinin, and 1 µg/ml leupeptin), after which 25 µl of lysozyme solution (stock: 10 mg/ml in Tris-HCl [pH 8.0]) were added. After mixing the samples by vortexing for 3 s, lysis was performed for 4 hr at 4°C with constant rotation. This was followed by three freeze/thaw cycles using liquid nitrogen and centrifugation for 10 min at 16,000 × *g* and 4°C. Soluble (supernatant) and insoluble (pellet) fractions were analysed separately by SDS-PAGE and Coomassie staining of the gels.

## Purification of inclusion bodies for antibody production

For large-scale purification of inclusion bodies, 1 l of an IPTG-induced culture was grown at 27°C until reaching an OD$_{600}$ 0.8 and harvested by centrifugation at 5000 rpm for 20 min. The pellet was washed once with 1× PBS and stored at −80°C until further processing. Lysis of the bacteria was performed by addition of 500 µl of bacterial lysis buffer and 50 µl of lysozyme solution (see above). Furthermore, the samples were mixed by vortexing for 3 s, incubated for 4 hr at 4°C with constant rotation, and subsequently subjected to three freeze/thaw cycles using liquid nitrogen. Inclusion bodies were pelleted by 20 min centrifugation at 15,000 rpm and 4°C, washed once with bacterial lysis buffer containing 1% Triton X-100, and eventually resuspended in 10 ml of 8 M urea. These were then left to dissolve for 16 hr at 4°C with constant rotation.

After determining the protein concentration spectrophotometrically (A280; Nanodrop, Thermo Fisher Scientific, Karlsruhe, Germany), 0.5 mg of the protein in inclusion bodies was then separated by SDS-PAGE and the band corresponding to NusA-CFB2 was cut from the gel. The gel slice was submitted to David's Biotechnology (Regensburg, Germany) for the generation of antisera in rabbit.

## Purification of anti-CFB2 antibodies

Polyclonal antisera were purified by adsorption of total rabbit serum to immobilized recombinant CFB2-Gb1 carrier protein. 100 µg of the CFB2 recombinant protein was resolved in SDS-PAGE gel and transferred to PVDF membrane. CFB2 containing region was excised, fragmented in small pieces, and transferred to a 1.5 ml tube. The membrane was blocked with 5% milk in PBS/0.05% Tween for 1 hr at room temperature with agitation, washed twice in PBS containing 0.05% Tween-20 and incubated with 1 ml of antisera for 16 hr at 4°C with agitation. After incubation, the depleted antiserum was discarded and the membrane washed three times in PBS/0.05% Tween-20. Finally, the bound antibodies were eluted with 200 µl of 0.1 M glycine, pH 2.4 for 5 min, under vortex and neutralized with 20 µl of Tris-HCl, pH 8.0.

## Tethering and yeast two-hybrid assays

Tethering assays were done using cells expressing mRNAs with a chloramphenicol acetyltransferase (CAT) open reading frame and a truncated version of the trypanosome *actinA* 3'-UTR, with five boxB

sequences immediately upstream of the 3'-UTR (*Mugo and Erben, 2020*). Lines with tetracycline-inducible expression of different versions of CFB2 (as shown in *Figure 7*) were made, and CAT was assayed with or without tetracycline addition. CAT activity was performed as previously described using 14C-labelled butyryl coenzyme A as the labelled substrate (*Mugo and Erben, 2020*). Yeast two-hybrid assays were done using the Matchmaker Yeast Two-Hybrid System (Clontech) following the manufacturer's recommendations. The DNAs of the protein ORFs were PCR-amplified and cloned into pGBKT7 or pGADT7. Subsequent mutations of the ORFs were achieved via site-directed mutagenesis (Q5 Site-Directed Mutagenesis, New England Biolabs). The plasmids were co-transformed pairwise into AH109 yeast strains, and co-transformants were selected on double drop-out medium (minimal SD media lacking Trp and Leu). Positive interactions were indicated by growth on quadruple drop-out medium (minimal SD media lacking Trp, Leu, His, and Ade). The interaction between murine p53 and SV40 large T-antigen served as the positive control, with LaminC and the SV40 large T-antigen as the negative bait (DNA-binding domain) and prey controls, respectively. Expression of bait and prey proteins was checked by detection of HA and c-myc epitopes, respectively.

## Acknowledgements

We thank J Gerst (Weizmann Institute) for providing the pcDNA4-MS2-CP-GFP-SBP plasmid and A Gkeka and F Aresta-Branco (DKFZ) for providing the cell line expressing the 3'-UTR of VSG2 and assistance in the generation of the tagged VSG construct. Medium was prepared by Ute Leibfried and Claudia Helbig (ZMBH). We thank Gloria Rudenko (Imperial College, London, UK) for the selectable VSG2 cell line and Kevin Leiss (ZMBH) for assistance in formatting additional genomes for use with Tryprnaseq. VSG 3'-UTR mutant cell lines were made and characterized by Meike Torwort and Lara Ruland during their BSc thesis work, and some CFB2 expression plasmids were made by Rachel Williams. We are indebted to Luise Krauth-Siegel (BZH) for allowing us to share her laboratory, including all equipment, after a disastrous flood in the ZMBH, and to Natalie Dirdjaja for helping us to move in. Electron microscopy was done by Charlotta Funaya of the Electron Microscopy Core Facility, Heidelberg University; mass spectrometry was done in ZMBH facility under the leadership of Thomas Ruppert, and RNA Sequencing was done by David Ibbersson of Bioquant, University of Heidelberg. We thank Frauke Melchior (ZMBH) and especially Georg Stoecklin (University of Mannheim and ZMBH) for useful discussions and suggestions. Grant applications by CC to support this project were rejected by the ERC and the DFG. This project was supported by CC's core funding from the University of Heidelberg (State of Baden-Württemberg) and by the ERC grant #649019 (RNAEDIT) to NP.

## Additional information

### Competing interests

Christine Clayton: Reviewing editor, *eLife*. The other authors declare that no competing interests exist.

### Funding

| Funder | Grant reference number | Author |
|---|---|---|
| H2020 European Research Council | 649019 | Nina Papavasiliou |
| University of Heidelberg | ND | Christine Clayton |

The funders had no role in study design, data collection and interpretation, or the decision to submit the work for publication.

### Author contributions

Larissa Melo do Nascimento, Conceptualization, Data curation, Formal analysis, Investigation, Methodology; Franziska Egler, Conceptualization, Formal analysis, Investigation; Katharina Arnold, Formal

analysis, Investigation; Nina Papavasiliou, Conceptualization, Resources, Formal analysis, Supervision, Funding acquisition, Methodology, Writing - original draft, Project administration, Writing - review and editing; Christine Clayton, Conceptualization, Resources, Software, Formal analysis, Supervision, Funding acquisition, Investigation, Visualization, Writing - original draft, Project administration, Writing - review and editing; Esteban Erben, Conceptualization, Data curation, Formal analysis, Supervision, Validation, Investigation, Visualization, Methodology, Writing - original draft, Writing - review and editing

### Author ORCIDs
Larissa Melo do Nascimento (iD) https://orcid.org/0000-0003-4359-3558
Christine Clayton (iD) http://orcid.org/0000-0002-6384-0731
Esteban Erben (iD) https://orcid.org/0000-0001-5179-7863

### Decision letter and Author response
Decision letter https://doi.org/10.7554/eLife.68136.sa1
Author response https://doi.org/10.7554/eLife.68136.sa2

## Additional files

### Supplementary files
• Supplementary file 1. Plasmids and oligonucleotides used in the experiments.

• Supplementary file 2. Mass spectrometry results. The legend is on the first sheet.

• Supplementary file 3. Effects of RNAi on the transcriptome: alignment to the Lister427_2018 genome. Reads were aligned to the Lister427_2018 genome. Each read was allowed to align once. The legend is on the first sheet of the table. Annotations are often approximate since it is sometimes not clear which TREU927 gene is a true homologue of the Lister427 versions, especially for gene families.

• Supplementary file 4. Effects of RNAi on the transcriptome: alignment to the TREU927 genome. Reads were aligned to the TREU927 genome. Each read was allowed to align once. The legend is on the first sheet of the table. The sheet that includes the unique gene set, in which each set of repeated genes has only one representative, allows judgement of whether any particular functional class is enriched.

• Transparent reporting form

### Data availability
The RNASeq raw data is available at Array Express with the accession number E-MTAB-9700. The proteomics data are available via ProteomeXchange with identifier PXD021772.

The following datasets were generated:

| Author(s) | Year | Dataset title | Dataset URL | Database and Identifier |
|---|---|---|---|---|
| Clayton C, Erben E | 2020 | Effect of CFB2 depletion in Trypanosoma brucei | https://www.ebi.ac.uk/arrayexpress/experiments/E-MTAB-9700/ | ArrayExpress, E-MTAB-9700 |
| Clayton C, Erben E | 2020 | The trypanosome Variant Surface Glycoprotein mRNA is stabilized by an essential unconventional RNA-binding protein | http://proteomecentral.proteomexchange.org/cgi/GetDataset?ID=PXD021772 | ProteomeXchange, PXD021772 |

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
