## [Decision Letter]

[Editors' note: this paper was reviewed by Review Commons.]

**Acceptance summary:**

This is a very nice study, which exploits the trypanosome system to investigate the role of mRNA-specific mRNPs in regulating protein (VSG) expression. The experimental approaches used are robust, the data clear and complete. The findings are of broad interest to anyone interested in understanding post-transcriptional mechanisms, as well as being of high interest to parasitologists

---

## [Author Response]

Reviewer #1 (Evidence, reproducibility and clarity (Required)):*Trypanosoma brucei* causes African sleeping sickness and related cattle disease, both diseases that urgently need new therapeutics. One reason for the lack of a drug or a vaccine is the parasite's way to escape the immune system: their cell surface is covered by the variant surface glycoprotein (VSG) of which many variants exist, but only one is expressed. The switching between the different VSG forms is called antigenic variation and involves a not fully understood epigenetic mechanism. It is essential for the parasite's survival that the VSG surface coat is very dense at any given time: antibodies of the host should not be able to recognise any invariant proteins on the cell surface that are 'hidden' in between the VSG molecules. Consequently, the VSG protein is the most abundant protein in the cell (10% of total). This high protein abundance is achieved by both transcriptional and posttranscriptional mechanisms. One major posttranscriptional mechanism is the stabilisation of the VSG mRNA. Two cis-elements in the VSG mRNA 3´UTR have been known for a long time to be essential for this stability (an 8-mer and a 16-mer). However, nothing was known about the underlying mechanism of VSG mRNA stabilisation. In this work, the authors have addressed this problem. They have purified the VSG mRNA from trypanosomes in two very different ways and, in both approaches, they found the cyclin Fbox protein 2 (CFB2) to co-purify. They have defined the full complex that binds to the VSG mRNA. Most importantly, the authors could clearly show the very specific effect on VSG mRNA stability when CFB2 was RNAi depleted. Moreover, CFB2 RNAi mostly phenocopied the phenotype that was previously described for VSG RNAi. The CFB2 protein is present in a very low copy number and the authors provide data suggesting that it may be tightly autoregulated by interaction with SKP1. The authors further show that the regulation of VSG mRNA stability by CFB2 depends on the 16-mer cis-element, but not on the 8-mer. The data are, throughout, very convincing, experiments are done with all the essential controls and the data are well presented. The conclusions are supported by the data. The authors have, beyond any doubt, finally identified the major posttranscriptional regulator protein that is responsible for VSG mRNA stability, a milestone in the field, and provide a mechanism on how it could work and be autoregulated. I only have one major point (and a few very minor points)My main criticism is on the introduction: major information is missing here or presented far too short. People from outside of the trypanosome field will find the paper almost impossible to understand. It is important to explain the life cycle and its stages (as these are mentioned later) as well as the parasites special transcription of mRNAs by PolI and PolII in more detail. Trypanosome translation initiation factors and PABPs should be introduced. Nomenclature of the VSG is also a confusing throughout. Why switching to VSG4 in Figure 8 for example. Also, it would be beneficial to phrase the question better and stress the importance of why this needs to be answered to understand the basic biology of the parasite.

We have extended the Introduction section as suggested. The reason for the switch is now explained.

Minor points:Line 76: ' supporting direct binding to mRNA in vivo'Is this true? I thought the poly(A) oligos can also purify protein complexes? (but I may be wrong)

Yes, but probably not very much when the complexes have been washed with lithium chloride and urea. In any case, the readers can find in the supplementary Table 1 the false discovery rate (FDR) values obtained for each identified protein for both purifications (oligodT and VSG/Tub antisense) taking into consideration the data from the control experiments.

Line 104: ‘Kinetoplastid specific’. Better ‘Trypanosome specific’ if its absent in Leishmania? The correlation between presence of antigenic variation and number of CFB could be worked out a little better, perhaps presented in a main Figure.

CFB genes are absent in Leishmania; thus, we have edited it as suggested. Since we do not actually know whether it has any biological meaning, we have also removed the association between the presence of multiple copies of CFB genes and antigenic variation.

Line 161: Tb927.8.1945, ad: 'encoding a hypothetical protein of unknown function'.

Done.

Line 202: MG132, better: 'the proteasome inhibitor MG132'

Done.

Line 310-311: no, best to delete this sentence.

We prefer to leave it in.

Reviewer #1 (Significance (Required)):There is no doubt about this being a truly significant contribution to the trypanosome field. Method-wise, it is also a nice example of how mRNA binding proteins can be identified and validated and there are clear mechanistic insights here into the regulation of the VSG mRNA. This is not frequently found, in any organism. I believe that this work will be publishable in any parasitology journal, and, once the introduction has been changed (see above) also in any RNA journal.Reviewer #2 (Evidence, reproducibility and clarity (Required)):The current study describes the isolation and characterisation of Variant SurfaceGlycoprotein (VSG) mRNA-bound proteins in the bloodstream form African trypanosome.CFB2 is identified as a VSG mRNA positive regulator which depends upon a conserved 16mer in the VSG mRNA 3'-UTR.1. The authors state in their abstract that "CFB2 is essential for VSG mRNA stability". They also "describe cis-acting elements within the VSG 3'-untranslated region that regulate the interaction". Expression of a GFP reporter appears to be reduced by only ~3-fold in bloodstream-form cells when the relevant cis-acting element (the 16mer) is removed, however (Figure 8B). This would suggest that the mRNA lacking the 16mer could still be relatively stable ("VSG mRNA is extremely stable, having a half-life of 1-2h compared with less than 20 min for most other mRNAs").Was half-life measured for an mRNA lacking a 16-mer or for VSG mRNA in cells lacking CFB2?

Yes, this was previously published, references are in the Introduction. The presence of the 16-mer in VSG is essential for survival in the *T. brucei* bloodstream stage (PMID: 28906055).

Could CFB2 impact mRNA maturation rather than stability?

The reporter experiments rule this out since in Kinetoplastids, the 3'-UTR sequence has no role in controlling polyadenylation, beyond a preference for sites with several A residues. This is now explicitly stated.

Also, which data demonstrate an altered interaction between CFB2 and the mRNA lacking a 16mer? The authors could consider adjusting these statements and also the quantitative impact that CFB2 has on VSG mRNA stability, as well as evidence supporting differing interactions between CFB2 and mRNAs containing or lacking the 16mer.

We do now show new data demonstrating that binding of CFB2 to the reporters depends on the VSG 3’-UTR and is unaffected by the 8-mer mutation. Unfortunately, the *GFPVSGm16mer* mRNA was too low in abundance to quantitate, even by qPCR. The 16-mer and 8-mer are the only sequences in the 3’-UTR that are conserved in different *VSG* mRNAs. Binding to the upstream UC-rich region remains a theoretical possibility but it seems very unlikely since this region is variable and such sequences are present in numerous other 3’-UTRs (for example, the α tubulin 3’-UTR, which is the first we looked at, includes the sequence CCUUCCUUCCCCUU). Our preliminary results suggest indeed that region is not involved (Suppl. Figure 13E). And in that case, why would mutating the 16-mer affect the response to CFB2 expression? We cannot rule out the possibility that CFB2 binds to m6A – it’s a chicken-and-egg problem, because mutation of the 16-mer eliminates the methylation. However, this too seems unlikely since m6A is by no means restricted to *VSG*
(*https://doi.org/10.1101/2020.01.30.925776*; PMID: 30573362). To find out it would be necessary to identify the m6A “writers”, and reduce their expression; this is well beyond the scope of this manuscript and is being actively pursued in another lab. An alternative would be to express soluble CFB2 for in vitro binding studies, but so far this has not been possible despite several attempts.

2. In relation to point 1 above, Figure 2A and Figure 3D show CFB2 binding to the VSG 3’-UTR, to the 16mer in the latter case. This interaction could be presented as a ‘model’ whereas it seems too speculative to be included in the current data-Figures. Indeed, the authors “suggest that CFB2 recognizes the 16mer” in their Discussion and do also consider alternatives.

A caveat has been added to the Figure 3D legend.

3. Given the emphasis on the experimental approach and "the potential to supply detailed biological insight into mRNA metabolism in any eukaryote" (end of abstract), can the authors explain how their method improves upon / differs from the approach of Theil et al., 2019 and other similar approaches?

Our approach is slightly different to the one described by Theil et al. (antisense oligo length, incubation temperature) and a detailed description of our protocol can be found in the Methods section. We have stressed the method because there is only one previous successful example attempting the purification of the protein bound to a native mRNA. Our intention is not to compare approaches but to encourage researchers willing to perform these experiments in a variety of other organisms.

Other points:1. Figure 2B: Why does N-GFP- SBP migrate more slowly in the Tet+ eluate? Also why does the slower-migrating form of the protein appear to dominate in Figure 2C?

N-GFP-SBP protein migrates as a single band. In Figure 2C, the membrane was first probed with anti-RBP10 and then with anti-GFP antibodies. What is observed in the input and flowthrough (I/FT) is RBP10 signal and not GFP. The concentration of N-GFP-SBP in the eluate is much higher than in the I/FT (it is the only protein visualized upon Ponceau staining in eluates). That causes the band to appear in the eluate as “ghost band” (ECL reagent is consumed in the middle region of the band) while in the I/FT, the concentration is still not enough to give a signal. The same occurs in Figure 2B. The faint bands that are seen in the I/FT in Figure 2B are likely products of cross-reactivity.

2. Figure 3D: What's the evidence that SKP1 interacts with VSG-mRNA-bound CFB2? Is this protein enriched in the data shown in Figure 1C and can the relevant data-point be labelled?

Our interactome capture results (PMID: 26784394) suggest that in bloodstream form, Skp1 (Tb927.11.6130) do not bind poly(A) RNA directly; thus, it is not enriched in the VSG mRNA-bound proteome. What we know is that Skp1 interacts, in a Y2H setting, with CFB2 and that mutations in the CFB2 F-box domain abolish this interaction. The data we have presented suggest the interaction with Skp1 regulates CFB2 levels. We actually do not know whether Skp1 binds to free or to VSG-mRNA-bound CFB2.

3. There are four other highly abundant mRNAs in Figure 4C. Are these related to VSG expression?

They are tubulins, EF1, HSP83 and HSP70.

4. Lines 85-88: Suggest citing the studies used to prioritise RBPs, expressed only in the bloodstream form, that increase mRNA stability or translation when "tethered" to an mRNA.

References have been added.

Is CFB2 expressed only in the bloodstream form?

Yes, this is described in more detail later.

5. We spotted a number of other potential corrections, including: Lines 161 and 171; should '4E' be '4C'? Line 202; explain MG132. Define RPM, ns, BS, ++ etc in the Figures. Yeast-2hybrid and CAT may be standard assays, but we suggest briefly describing them in the Methods section.

Done.

Reviewer #2 (Significance (Required)):Post-transcriptional control of gene expression by mRNA binding proteins (RBPs) is an area of major current research interest and activity. Much remains unknown regarding control of mRNA stability, nuclear export or translation and there are many uncharacterised or only partially characterised RBPs in eukaryotic cells. Trypanosomes present an important model in this context since global polycistronic transcription places a major emphasis on posttranscriptional controls. They are also important parasites. The variant surface glycoprotein is a key virulence factor and one of the few genes that is under transcriptional control in African trypanosomes, yet RBPs are thought to be important for generating/maintaining the highly abundant VSG mRNA in bloodstream form cells (and for low abundance in the insect stage), possibly via interaction with the highly conserved regulatory elements in the 3'-UTR.